# What is Wrong with Perplexity for Long-context Language Modeling?

**Lizhe Fang**[1][*]  **Yifei Wang**[2][*]  **Zhaoyang Liu**[3]  **Chenheng Zhang**[1]
**Stefanie Jegelka**[4,5]  **Jinyang Gao**[3]  **Bolin Ding**[3]  **Yisen Wang**[1,6][†]

[1] State Key Lab of General Artificial Intelligence,
   School of Intelligence Science and Technology, Peking University
[2] MIT CSAIL
[3] Alibaba Group
[4] TUM CIT, MCML, MDSI
[5] MIT EECS, CSAIL
[6] Institute for Artificial Intelligence, Peking University

## Abstract

Handling long-context inputs is crucial for large language models (LLMs) in tasks such as extended conversations, document summarization, and many-shot in-context learning. While recent approaches have extended the context windows of LLMs and employed perplexity (PPL) as a standard evaluation metric, PPL has proven unreliable for assessing long-context capabilities. The underlying cause of this limitation has remained unclear. In this work, we provide a comprehensive explanation for this issue. We find that PPL overlooks key tokens, which are essential for long-context understanding, by averaging across all tokens and thereby obscuring the true performance of models in long-context scenarios. To address this, we propose **LongPPL**, a novel metric that focuses on key tokens by employing a long-short context contrastive method to identify them. Our experiments demonstrate that LongPPL strongly correlates with performance on various long-context benchmarks (e.g., Pearson correlation of -0.96), significantly outperforming traditional PPL in predictive accuracy. Additionally, we introduce **LongCE** (Long-context Cross-Entropy) loss, a re-weighting strategy for fine-tuning that prioritizes key tokens, leading to consistent improvements across diverse benchmarks. These contributions offer deeper insights into the limitations of PPL and present effective solutions for accurately evaluating and enhancing the long-context capabilities of LLMs. Code is available at https://github.com/PKU-ML/LongPPL.

## 1 Introduction

The ability to process long-context inputs is critical for large language models (LLMs) in many real-world tasks, such as long conversations (Maharana et al., 2024), document summarization (Chang et al., 2024), and many-shot in-context learning (Agarwal et al., 2024; Li et al., 2024; Wei et al., 2023). Despite many techniques for extending the context length (Han et al., 2023; Chen et al., 2023; Zhu et al., 2024; Xiong et al., 2024; Chen et al., 2024a), the evaluation of long-context capabilities still widely uses perplexity (PPL) as the *de facto* metric. Many have claimed to extend context windows to 32k, 128k, or even millions of tokens, based on attaining a low perplexity score under long context. However, recent studies have challenged this common practice by revealing a huge discrepancy between perplexity and actual performance on long-context tasks (Hu et al., 2024a; Hsieh et al., 2024). As shown in Figure 1(b) (top), the perplexity of LLMs shows almost no correlation to their long-context performance measured by Longbench scores (Bai et al., 2023b). This raises the question:

*Why does perplexity fail to reflect the long-context abilities of LLMs?*

---

[*]Equal Contribution.
[†]Corresponding Author: Yisen Wang (yisen.wang@pku.edu.cn).

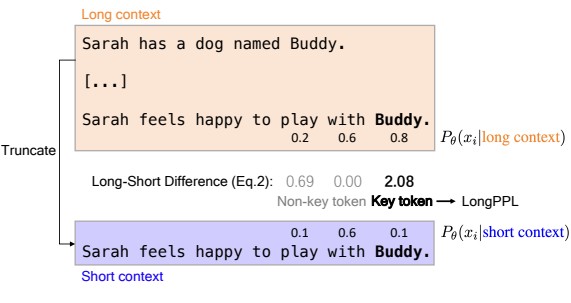

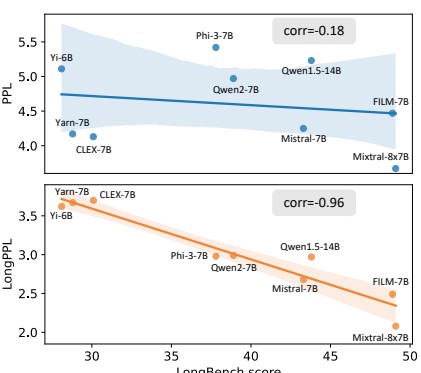

(a) Illustration of how LongPPL is calculated.

(b) LongBench *vs.* PPL / LongPPL (Ours)

Figure 1: **(a)** A constructed example to illustrate how LongPPL is calculated. We truncate the long context and calculate the generation probability difference (long-short difference, LSD, Eq. (2)) for each token based on the long and short contexts. A high LSD score indicates that the token's generation is significantly enhanced by the long context, making it a key token in the long text. LongPPL is then obtained by calculating perplexity on these key tokens. **(b)** Long-context performance (Long-Bench (Bai et al., 2023b)) *vs.* perplexity measures (PPL and our LongPPL) computed on GovReport (Huang et al., 2021), a natural corpus. While PPL shows no correlation *w.r.t.* Longbench score, LongPPL achieves $-0.96$ Pearson correlation coefficient.

To understand this phenomenon, we conduct a fine-grained analysis of the roles of different tokens at long-context tasks. Notably, we find perplexity computed only on the answer tokens to the long-context tasks strongly correlates with LongEval accuracy, whereas perplexity on non-answer tokens shows little to no correlation. Since most tokens are non-answer tokens, standard perplexity averaging over all token *equally* fails to represent the long-context abilities. This motivates us to average over the *key tokens* that reflect a model's long-context abilities. A key obstacle is that natural texts have no ground-truth reference of key tokens, making it hardly applicable to general cases.

To tackle this challenge, we propose a principled method to measure the influence of long context on each token by performing a causal intervention on its context length. We find that tokens with significantly better predictions under long context are strongly tied to long-context information, even though they make up only a small portion of general text. Empirically, our proposed method can accurately identify the answer tokens in LongEval with up to 98.2% accuracy.

Built upon the accurate selection of key tokens, we propose **LongPPL** (Long-context Perplexity), where we compute perplexity by only averaging solely on the selected key tokens (Figure 1(a)). Extensive experiments across a diverse suite of LLMs and long-context benchmarks show that LongPPL computed on natural language corpus exhibits a consistently strong correlation with their benchmark scores computed over various long-context tasks, *e.g.,* -0.96 correlation in Figure 1(b) (bottom). Thus, LongPPL offers a natural way to evaluate LLMs' long-context capabilities in an *unsupervised* fashion.

Following the design of LongPPL, we further develop an efficient long-context training strategy by *emphasizing key tokens*. Specifically, we propose the **LongCE** (Long-context Cross-Entropy) loss that upweights the key tokens, which can be estimated by the model itself. In this way, LongCE can bootstrap its long-context abilities by alternating between estimating key tokens and optimizing key tokens. Experimental results across multiple LLMs show that LongCE consistently improves over the conventional CE loss, with a maximum accuracy gain of 22% on LongEval.

Our contributions are summarized as follows:

- We conduct a fine-grained analysis on the failure of perplexity at measuring long-context abilities. Specifically, we reveal the critical roles of key tokens in long-context tasks and propose principled metrics to identify key tokens with high accuracy.

- We propose **LongPPL** (Long-context Perplexity) that is solely based on the selected key tokens. Extensive evaluation shows that in contrast to standard PPL, LongPPL exhibits a strong correlation with long-context abilities across multiple LLMs and benchmarks.
- We introduce **LongCE** (Long-context Cross Entropy) loss that assigns larger weights to key tokens that gain more from the long context. LongCE attains consistent improvements in a plug-and-play solution, demonstrating its generality for learning long-context models.

## 2 A FINE-GRAINED ANALYSIS OF PERPLEXITY

Recent studies have shown that perplexity does not adequately reflect the long-context performance of language models (Agarwal et al., 2024; Li et al., 2024), as we have also observed in Figure 1(b). In this section, we demystify this phenomenon with a fine-grained analysis of the roles of different tokens at long-context performance.

Perplexity is a commonly used metric for evaluating a LM's ability to predict the next word in a sequence (Jelinek et al., 1977). For a sequence of tokens $\boldsymbol{x} = (x_1, x_2, ..., x_n)$, a language model parameterized by $\theta$ is learned to predict the conditional probability of each token given the previous context $P_\theta(x_i|\boldsymbol{x}_{<i}), i \in [n]$. The perplexity (PPL) on this sequence is defined as the inverse of the geometric mean of all token probabilities:

$$\mathrm{PPL}_\theta(\boldsymbol{x}) = \exp\left(-\frac{1}{n}\sum_{i=1}^{n}\log P_\theta(x_i|\boldsymbol{x}_{<i})\right) = P_\theta(\boldsymbol{x})^{-\frac{1}{n}}. \tag{1}$$

It quantifies the model's uncertainty when encountering new tokens. A larger likelihood of $\boldsymbol{x}$ indicates better prediction and lower perplexity.

### 2.1 NOT ALL TOKENS MATTER FOR LONG-CONTEXT PERFORMANCE

Despite the close connection between perplexity and token prediction accuracy, there is growing evidence that LLMs' perplexity does not indicate their performance on long-context benchmarks (Hu et al., 2024a; Hsieh et al., 2024). There are two possible sources of this mismatch: either the log-likelihood-based metric is flawed, or the averaged tokens are not representative enough. In this work, we *champion the latter explanation* by showing that when selecting the proper "key tokens" for long-context understanding, perplexity can correlate very well with long-context performance.

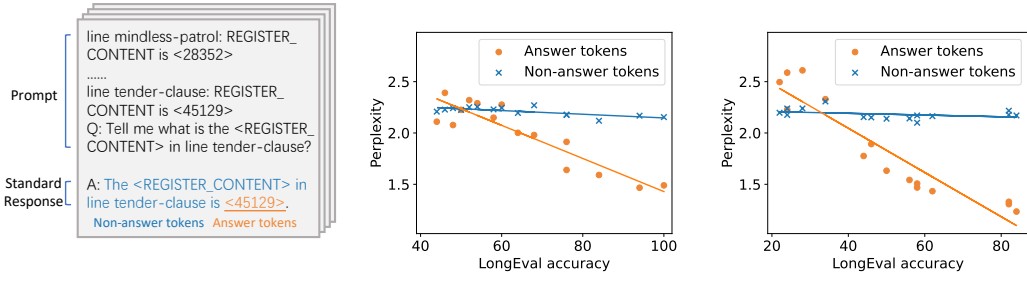

(a) Example of answer tokens. (b) PPL vs LongEval (Yi-6B) (c) PPL vs LongEval (CLEX-7B)

Figure 2: **(a)** An example of the answer tokens in the LongEval task. **(b&c)** The correlation between accuracy and perplexity on answer tokens / non-answer tokens on LongEval. Each point represents the results obtained from testing at a specific prompt length ranging from 2k to 28k. The experiments is conducted using Yi-6B-200K (Young et al., 2024) and CLEX-7B-64K (Chen et al., 2024a).

To have an intuitive understanding, let us consider a real example from LongEval benchmark shown in Figure 2(a). Most tokens in the answer, "the <REGISTER_CONTENT> in line tender-clause is", are straightforward answer formats stemmed immediately from the question, without relying on any long-context information. Even short-context LLMs can predict well on these tokens. Since most tokens are *long-context-agnostic* tokens, perplexity computed *equally* over all tokens do not represent long-context performance.

To quantitatively examine this hypothesis, we conduct experiments on LongEval (Li et al., 2023a), a benchmark for long-context retrieval abilities, where we can separate the *answer tokens* that match

the desired answers (*e.g.,* <45129> in Figure 2(a)) from *non-answer tokens*. We compare the perplexity computed with these two groups of tokens using two long-context LLMs. As shown in Figures 2(b) & 2(c) (result details in Appendix B.4), the perplexity on answer tokens correlates strongly with the LongEval accuracy that represents the long-context performance; instead, the perplexity on the non-answer tokens shows almost no correlation with LongEval accuracy, justifying our intuition that these tokens do not matter for evaluating long-context performance. In other words, we should evaluate the perplexity of the key tokens that really matter for long-context performance.

## 2.2 EXTRACTING KEY TOKENS FROM NATURAL TEXTS

In natural texts used for training LLMs, we do not have knowledge of the answer tokens as in LongEval experiments (Figure 2). This motivates us to find a surrogate metric that can accurately identify the key tokens that matter for long-context performance.

To measure the influence of long context for each token $x_i$, we perform an *intervention* of context length. Specifically, given a sequence $x$ and a language model $P_\theta$ (with strong long-context abilities), for each token $x_i$ that has a long context, we compute the difference between its log probability under the full *long context* $l_i = (x_1, \ldots, x_{i-1})$ and the log probability under the truncated *short context* $s_i = (x_{i-K}, \ldots, x_{i-1})$ (where $K$ is a short length, *e.g.,* 64):

$$\text{LSD}_\theta(x_i) = \log P_\theta(x_i|l_i) - \log P_\theta(x_i|s_i). \tag{2}$$

We call it **Long-Short Difference (LSD)**, which measures the improvement in prediction accuracy *endowed solely by the long context*. From a causal perspective, $s_i$ serves as the counterfactual context created by the intervention (dropping long context), and the LSD estimates the individual treatment effect (ITE) (Hernán & Robins, 2010) of long context using the language model $P_\theta$. Thus, a high LSD value indicates that long context plays an important part in the prediction of $x_i$, making them the key tokens to be considered for evaluating long-context performance. In other words, LLMs good at long-context understanding should be able to predict high-LSD tokens accurately.

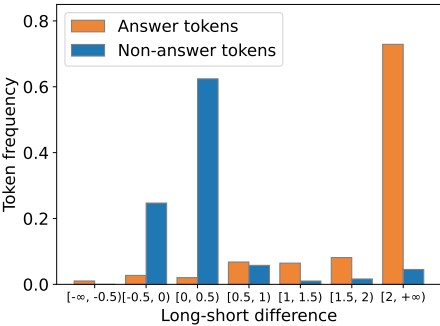
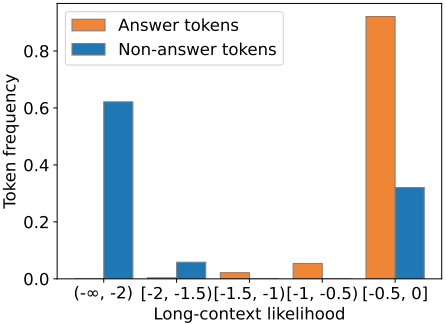

(a) LSD of tokens on LongEval.  (b) LCL of tokens on LongEval with large LSD.

Figure 3: **(a)** Token distribution categorized by long-short difference (LSD). **(b)** Distribution of tokens with LSD greater than 0.5 categorized by long-context likelihood (LCL). The tokens are from the standard response of LongEval illustrated in Figure 2(a).

We evaluate the LSD score on LongEval, where we have knowledge of the key answer tokens. As shown in Figure 3(a), we compute the LSD score with a powerful long-context LLM, Mixtral-8x7B (Jiang et al., 2024), and find that answer tokens are clearly separated from the non-answer tokens: most answer tokens have LSD values higher than 2, while most of the non-answer tokens concentrate around low LSD values (lower than 0.5). When using LSD values alone to classify answer and non-answer tokens, we attain 85.6% accuracy (Figure 4(b)), indicating that LSD values are strongly indicative of the key tokens in long-context understandings.

From Figure 3(a), we find that a small proportion of non-answer tokens also have large LSDs (larger than 0.5) and are thus confused together with key tokens. After analyzing, we find that these tokens can be further separated out by inspecting their **Long-Context Likelihood (LCL)** under long context:

$$\text{LCL}_\theta(x_i) = \log P_\theta(x_i|l_i) = \log P_\theta(x_i|x_{<i}). \tag{3}$$

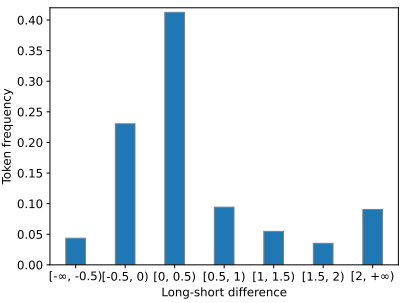

(a) LSD value distribution on GovReport.

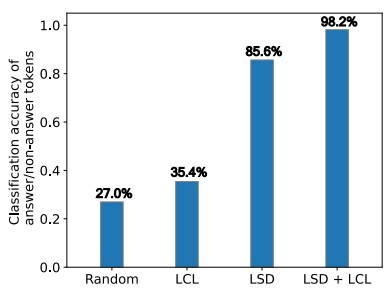

(b) Criteria to identify answer tokens.

Figure 4: **(a)** Distribution of tokens in GovReport categorized by long-short difference. **(b)** The classification accuracy of discriminating answer to non-answer tokens on LongEval with a classifier using different metrics (Random refers to a 50-50 random guess on two classes).

A lower LCL indicates that the language model hardly predicts accurately at $x_i$ even with the long context information. Figure 3(b) shows that these high-LSD non-answer tokens actually have lower LCLs than the corresponding answer tokens, indicating that these tokens are (strongly) mispredicted tokens even under a long context. In other words, these tokens are fundamentally hard to predict regardless of the context. Therefore, we can exclude them from the selection of key tokens.

To summarize, we revisit our initial question why perplexity fails to represent long-context performance. As shown in Figure 4(a), most tokens in a natural corpus, GovReport (Huang et al., 2021), are long-context-irrelevant tokens with low LSD (lower than 0.5), while only less than 10% tokens are highly influenced by long context (with LSD> 2) and represent long-context abilities. Therefore, perplexity that averages over all tokens (Equation 1) does not represent the real long-context performance. Instead, combining the LSD (Equation 2) and the LCL (Equation 3) scores, we are able to accurately identify the answer tokens in LongEval with an accuracy of **98.2%** (Figure 4(b)). Based on this result, in the next section, we design a new perplexity measure, LongPPL, that is tailored to reflect the long-context performance of LMs, by focusing on the key tokens.

## 3 MEASURING AND ENHANCING LONG-CONTEXT CAPABILITIES WITH KEY TOKENS

In Section 2, we find that only key tokens correlate well with long-context performance (Section 2.1), and we identify two effective measures to select the key tokens from a natural corpus (Section 2.2). Based on these observations, we design a new perplexity measure, LongPPL, to measure the long-context abilities, and, following in the same vein, we propose a new training objective, LongCE, for finetuning LLMs with an emphasis on key tokens.

### 3.1 LONG-CONTEXT PERPLEXITY (LONGPPL)

Given a sequence $\boldsymbol{x} = (x_1, \ldots, x_n)$ and a language model $P_\theta$ to be evaluated, we consider a generalized notion of perplexity for long context understanding, Long-context Perplexity (LongPPL), where we can assign an influence function $I(\cdot) : \mathbb{X} \to \mathbb{R}_+$ to each token $x_i$:

$$\text{LongPPL}(\boldsymbol{x}; \theta, \theta_0) = \exp\left(\sum_{i=1}^n -\hat{I}(x_i; \theta_0) \log P_\theta(x_i | \boldsymbol{x}_{<i})\right),$$

$$\text{where } I(x_i; \theta_0) = \begin{cases} 1, & \text{if } \text{LSD}_{\theta_0}(x_i) > \alpha \text{ and } \text{LCL}_{\theta_0}(x) > \beta; \\ 0, & \text{otherwise.} \end{cases} \qquad (4)$$

$$\text{and } \hat{I}(x_i) = I(x_i) / \sum_j I(x_j).$$

Here, the *long-context influence* of $x_i$, $I(x_i; \theta_0) \geq 0$, selects key tokens to have a large long-short difference (LSD, Equation 2) and a large long-context likelihood (LCL, Equation 3) based on

an evaluator model with parameters $\theta_0$, with two threshold parameters $\alpha, \beta$. $\hat{I}(x_i)$ is the relative influence after normalization. The first criterion ensures that the generation of the token is enhanced by the additional information in the long-context. The second criterion excludes the fundamentally hard (misclassified) tokens that long context information does not help. Based on these criteria, all tokens are divided into two categories. Tokens that meet the criteria are selected as key tokens and are included in the perplexity calculation with equal weight, while those that do not meet the criteria are excluded from the calculation. Later in Section 4.1, we show that in contrast to standard PPL, LongPPL computed on a natural language corpus for multiple LLMs correlates well with their performance on long-context benchmarks, including LongEval (Li et al., 2023a), LongBench (Bai et al., 2023b), and RULER (Hsieh et al., 2024). We also consider other similar variants of the influence function (*e.g.,* with soft reweighting) and find them to be generally effective (though often less accurate).

**Remark on the Evaluator Model $\theta_0$.** Notably, the evaluator $P_{\theta_0}$ used for computing the long-context influence **can be different from the evaluated model** $P_\theta$. In fact, for the evaluator, we need a powerful model to ensure that they give a relatively accurate estimate of the token's long-context influence. This requires the evaluator itself to have a strong long-context understanding ability. Our empirical findings show that using the model $P_\theta$ itself as the evaluator $P_{\theta_0}$ leads to LongPPL being unable to distinguish the model's long-context capabilities (Appendix B.2). In practice, we find that a small-sized model like Llama-3.1-8B (Dubey et al., 2024) is enough to serve as a good evaluator.

## 3.2 IMPROVING LONG-CONTEXT CAPABILITIES WITH LONGCE

Due to the massive computational cost of pre-training an LLM from scratch on long texts, current long-context LLMs are pretrained on short contexts and then fine-tuned on longer contexts. By default, the long-context fine-tuning process adopts the Cross Entropy (CE) loss as in pre-training, which adopts a uniform average of all tokens, akin to standard perplexity (Equation 1):

$$\text{CE}(x; \theta) = -\frac{1}{n} \sum_{i=1}^{n} \log P_\theta(x_i | \boldsymbol{x}_{<i}). \tag{5}$$

Nevertheless, this *de facto* paradigm has the same issues that we discussed for perplexity in Section 2. We show that most tokens in a sequence are not influenced by the long context, while only a few key tokens require long-context information; and in turn, the model's long-context performance depends crucially on its prediction on these key tokens (as measured in LongPPL, Section 3.1).

Following the methodology of LongPPL (Equation 4), we propose the **LongCE** (Long-context Cross Entropy) loss that reweights every token $x_i$ *w.r.t.* its gain $I_{\text{soft}}(x_i; \theta)$ from long context:

$$\text{LongCE}(x; \theta) = -\frac{1}{n} \sum_{i=1}^{n} I_{\text{soft}}(x_i; \theta) \log P_\theta(x_i | \boldsymbol{x}_{<i}). \tag{6}$$

For the ease of differentiable optimization using all tokens, we adopt a *soft* long-context influence function $I_{\text{soft}} : \mathbb{X} \to [0, \gamma]$ based on the likelihood ratio between the long-context probability $P_\theta(x_i | \boldsymbol{l}_i)$ and short-context probability $P_\theta(x_i | \boldsymbol{s}_i)$ (defined in Section 2.2):

$$I_{\text{soft}}(x_i; \theta) = \min\left(\exp\left(\text{LSD}_\theta(x_i)\right), \gamma\right) = \min\left(\frac{P_\theta(x_i | \boldsymbol{l}_i)}{P_\theta(x_i | \boldsymbol{s}_i)}, \gamma\right). \tag{7}$$

Here, $\gamma > 0$ is a hyper-parameter that sets a threshold on the maximal influence to avoid numerical instability. As a consequence of this reweighting term, too easy tokens (both short and long context give accurate prediction) and too hard tokens (neither short or long context predicts correctly) will have a weight around 1, while those long-context-dependent tokens (high $P_\theta(x_i | \boldsymbol{l}_i)$ and low $P_\theta(x_i | \boldsymbol{s}_i)$) will be upweighted above 1, proportionally to the context informativeness.

**Remark.** Unlike the influence function of LongPPL (Equation 4), which uses a powerful LLM as an external evaluator to select tokens more effectively, LongCE leverages the **same model to evaluate the influence** for training efficiency. Therefore, LongCE training does not require a separate evaluator model, but uses the model itself for long-context evaluation. In this way, *LongCE bootstraps the model's long-context capabilities in an EM (expectation-maximization) way*: the language model $P_\theta$ first uses itself to estimate long-context influence of each token $I_{\text{soft}}$ (Equation 7); and then this estimate is used to update the model parameters by optimizing the LongCE loss function LongCE

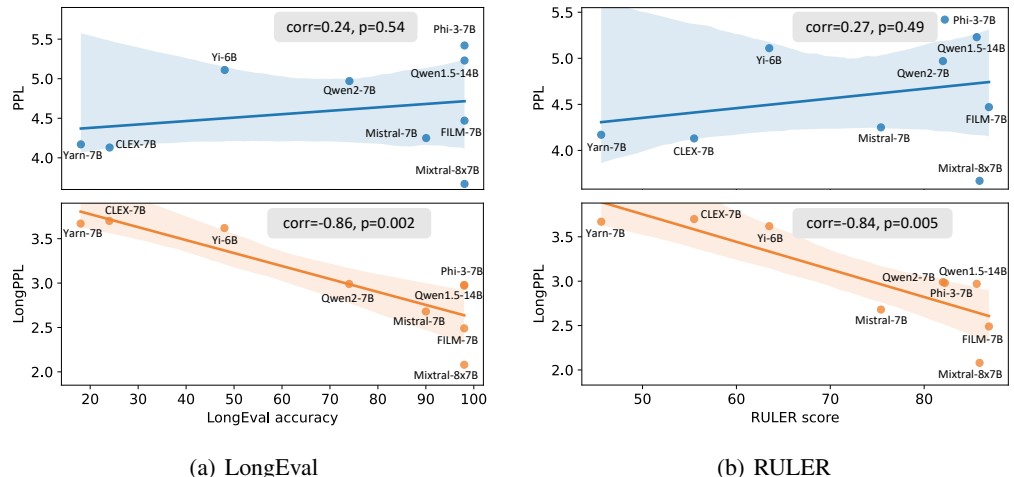

Figure 5: Correlation between the PPL-based metrics (LongPPL and PPL) on GovReport (Huang et al., 2021) and long-context benchmarks. LongPPL is calculated using Qwen2-72B-Instruct. Results of LongBench is in Figure 1(b).

(Equation 6). This process enables the model to focus more effectively on the key tokens critical to long-context performance, thereby improving training efficiency. We also note that computing key tokens introduces some additional computational overhead. However, subsequent experiments show that this overhead is acceptable, given the clear performance improvements.

## 4 EXPERIMENTS

In this section, we conduct real-world experiments to analyze the applicability of the proposed LongPPL and LongCE. For all the experiments, we use LongBench (Bai et al., 2023b), LongEval (Li et al., 2023a), and RULER (Hsieh et al., 2024) as the long-context benchmarks. We report the average score on LongBench, the accuracy on the subtask "lines" of LongEval, and the score on RULER. For LongBench and RULER, we restrict the prompt length to 32k tokens. For LongEval, we use 1350 lines as the prompt, which is approximately 32k tokens.

**Practical Implementation.** In the implementation of LongPPL and LongCE, we need to compute the log probabilities for each token under both the long and the truncated short context. For the truncated short context of length $K$, one can use the sliding window technique in Transformers for computing token predictions in parallel to improve computational efficiency. For computing LongPPL when the evaluator model and the evaluated model have different tokenizers, we only keep key tokens that form the longest common substrings of the evaluated tokens. More details can be found in Appendix A.1.

### 4.1 LONGPPL METRIC

**Experimental Setup.** We calculate LongPPL on the GovReport dataset (Huang et al., 2021), which consists of long sequences from government reports. We sample 50 documents with the context length up to 32k tokens. We set the hyperparameters as $\alpha = 2, \beta = -2, K = 4096$. We use Qwen2-72B-Instruct (Yang et al., 2024), an open-source LLM with the context length of 128k tokens, as the discriminator model $\theta_0$ to select the key tokens. We also consider using Llama-3.1-8B (Dubey et al., 2024) later and Mistral Large 2 (Jiang et al., 2023) in Appendix B.1.

**LongPPL Correlates Well with Long-context Performance.** In Figure 1(b) and Figure 5, we demonstrate the correlation between LongPPL and long-context benchmarks on various long-context LLMs. We observe that LongPPL exhibits a very strong negative correlation with performance on long-context tasks across different models, with pearson correlation coefficients exceeding

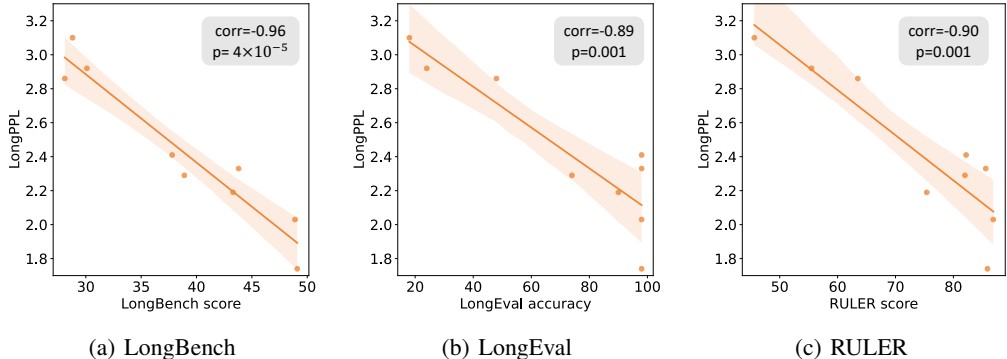

(a) LongBench       (b) LongEval       (c) RULER

Figure 6: Correlation between LongPPL on GovReport and long-context benchmarks. LongPPL is calculated using Llama-3.1-8B.

Table 1: The Pearson correlation between different perplexity measures and benchmark scores, where a lower correlation is the better (since we expect a lower perplexity indicates higher benchmark scores).

| Metrics | Influence $I$ | LongBench | LongEval | RULER |
|---|---|---|---|---|
| PPL | $I(x) \equiv 1$ | -0.11 | 0.31 | 0.33 |
| LongPPL-soft | $I_{\text{soft}}$ (Equation 7) | -0.43 | -0.21 | -0.17 |
| LongPPL-hard (default) | $I$ (Equation 4) | **-0.96** | **-0.86** | **-0.84** |

-0.8 for all three tasks. In contrast, perplexity hardly shows a correlation with the long-context tasks. This indicates that LongPPL is sufficiently capable of measuring a model's long-context capabilities.

**LongPPL is Compatible with Small-sized Evaluator Models.** To demonstrate that the effectiveness of LongPPL is not restricted by the size of the evaluator model, we additionally conduct experiments on a smaller model, Llama-3.1-8B (Dubey et al., 2024). As shown in Figure 6, the LongPPL computed using an 8B-sized model also achieves high correlation coefficients of -0.96, -0.89, and -0.90 with the three long-context benchmarks, respectively. In Appendix B.8, we have made discussion about the efficiency of LongPPL.

**Hard Standard for Key Tokens is Better than Soft Re-weighting Standard.** In Equation 4, we use an indicator function $I$ as the influence function. Instead, we have also tried to use the soft reweighting function $I_{\text{soft}}$ used in LongCE (Equation 7) to calculate LongPPL. Its token matching strategy is detailed in Appendix A.1. In Table 1, we show that LongPPL with soft criteria has a weaker correlation with the long-context benchmarks compared to LongPPL, indicating that the soft reweighting influence function is suboptimal for LongPPL. Besides, in Appendix B.2 and B.7, we have also explored some other alternative approaches, including using the model itself as the evaluator, removing the LCL discriminative condition, and using N-grams as the key token discriminative condition. We find that all of these approaches led to worse performance.

**LongPPL is not sensitive to the choice of hyperparameters of $\alpha$ and $\beta$.** To investigate the impact of the two threshold hyperparameters, *i.e.*, $\alpha$ and $\beta$ (in Equation 4), we conducted further ablation experiments. The results are presented in Table 2. Our findings reveal that when $\beta$=-1, $\alpha$=1 or 2, the correlation between LongPPL and the long-context benchmarks even improves. Notably, these hyperparameters were directly reused from the motivation experiments without any further tuning. The results indicate that LongPPL's performance is largely insensitive to the choice of hyperparameters, with the correlation coefficient remaining below -0.8 in most cases.

## 4.2 FINE-TUNE WITH LONGCE LOSS

**Experimental Setup.** We primarily use Llama-2-7B (Touvron et al., 2023) as the base model to perform long-context finetuning. We also conduct experiments on Mistral-7B-v0.1 (Jiang et al., 2023) and Llama-2-13B. We use PG-19 (Rae et al., 2020), a book dataset sourced from a library,

Table 2: The Pearson correlation between LongPPL, calculated with different hyperparameters ($\alpha$, $\beta$), and the long-context benchmarks. In most cases, the correlation coefficients remain below -0.8.

| LongPPL | LongBench | LongEval | RULER |
|---|---|---|---|
| $\alpha = 2, \beta = -2$ (default) | -0.96 | -0.86 | -0.84 |
| $\alpha = 2, \beta = -1$ | -0.96 | **-0.92** | **-0.92** |
| $\alpha = 1, \beta = -2$ | -0.91 | -0.73 | -0.69 |
| $\alpha = 1, \beta = -1$ | **-0.97** | -0.88 | -0.87 |

Table 3: Long-context performance of the fine-tuned models using the standard CE loss and our proposed LongCE loss. We fine-tune Llama-2-7b on long texts using various fine-tuning strategies (EABF and PI) and different training data (PG-19 and Pile-arxiv). The models are then assessed on benchmarks with prompts of up to 32k tokens.

| | LongBench | | | LongEval | | | RULER | | |
|---|---|---|---|---|---|---|---|---|---|
| Training steps | 50 | 100 | 200 | 50 | 100 | 200 | 50 | 100 | 200 |
| *Setting A (PG-19 dataset with EABF)* | | | | | | | | | |
| CE | 24.5 | 26.6 | 26.9 | 16.0 | 24.0 | 24.0 | 34.5 | 38.6 | 42.7 |
| LongCE (Ours) | **26.0** | **27.2** | **28.2** | **24.0** | **46.0** | **46.0** | **43.1** | **48.3** | **49.7** |
| **Gain** | (+1.5) | (+0.6) | (+1.3) | (+8.0) | (+22.0) | (+22.0) | (+8.6) | (+9.7) | (+7.0) |
| *Setting B (PG-19 dataset with PI)* | | | | | | | | | |
| CE | 24.3 | **25.3** | 25.4 | 20.0 | 28.0 | 26.0 | 22.1 | 31.8 | 35.7 |
| LongCE (Ours) | **24.4** | 25.0 | **25.8** | **38.0** | **44.0** | **42.0** | **27.3** | **34.4** | **36.4** |
| **Gain** | (+0.1) | (-0.3) | (+0.4) | (+18.0) | (+16.0) | (+16.0) | (+5.2) | (+2.6) | (+0.7) |
| *Setting C (Pile-arxiv dataset with EABF)* | | | | | | | | | |
| CE | 15.0 | 23.1 | 23.8 | 8.0 | **18.0** | 14.0 | 40.9 | 53.3 | 51.9 |
| LongCE (Ours) | **17.6** | **24.0** | **25.0** | **10.0** | **18.0** | **16.0** | **49.7** | **54.8** | **58.6** |
| **Gain** | (+2.6) | (+0.9) | (+1.2) | (+2.0) | (+0.0) | (+2.0) | (+8.8) | (+1.5) | (+6.7) |

and Pile-arxiv (Gao et al., 2020), a dataset consisting of Arxiv papers, as the training dataset. The training sequences are organized to be the context length with 32k tokens. For the calculation of LongCE, we set $\gamma = 5$ in Equation 7 and use the same sliding window approach as described in Section 4.1 to improve training efficiency. The context length of $s_i$ is set to be $K = 4096$. We fine-tune the base models with Entropy-aware Adjusted Base Frequency (EABF) (Zhang et al., 2024c) and Position Interpolation (PI) (Chen et al., 2023). Specifically, EABF applies a scaling mechanism to the attention and uses a higher base frequency for RoPE, while PI linearly downscales the position indices of the input tokens. These methods can significantly accelerate the convergence speed of long-context fine-tuning and have been widely adopted in many LLMs (Yang et al., 2024; Dubey et al., 2024; Chen et al., 2024a). Detailed training setups are available in Appendix A.2.

**LongCE Outperforms CE in Various Settings.** As shown in Table 3, we present the long-context capabilities of models fine-tuned with LongCE loss and CE loss under different fine-tuning strategies and training datasets (see fine-grained results of LongBench in Appendix B.3). We also test the effectiveness of LongCE using different base models in Table 4. We find that models fine-tuned with LongCE loss consistently outperform those fine-tuned with CE loss across nearly all settings. This suggests that the LongCE loss, with its re-weighting strategy based on long-context token importance, can be applied as a plug-and-play module which can effectively improve the model's long-context performance. To demonstrate the model's performance when the context length is over 32K, we provide the Needle-in-a-Haystack (Kamradt, 2023) evaluation results in Appendix B.5, which leads to similar conclusions. Besides, empirical results in Appendix B.6 demonstrate that LongCE does not cause any additional loss in the model's performance on normal-length tasks.

**Training Efficiency.** In addition to the performance improvement brought by the LongCE loss, we also pay attention to the changes in training efficiency. In LongCE, we need an extra forward pass to calculate the probability under short context $P_\theta(x_i|s_i)$, which introduces additional computation costs. By using a sliding window technique (as detailed in Appendix A.1), the computational overhead of training the model with LongCE is controlled to about 80% that of training with CE loss.

Table 4: Long-context performance of different fine-tuned models. We fine-tune Mistral-7B-v0.1 and Llama-2-13B with EABF adjustment strategy on Pile-arxiv dataset.

| Training steps | LongBench | | | LongEval | | | RULER | | |
|---|---|---|---|---|---|---|---|---|---|
| | 50 | 100 | 200 | 50 | 100 | 200 | 50 | 100 | 200 |
| *Mistral-7B-v0.1* | | | | | | | | | |
| CE | 29.6 | 28.9 | 28.4 | 26.0 | 14.0 | 12.0 | 45.0 | **44.5** | 42.9 |
| LongCE (Ours) | **30.8** | **30.9** | **31.1** | **36.0** | **30.0** | **26.0** | **45.1** | 44.0 | **43.5** |
| **Gain** | (+0.8) | (+2.0) | (+2.7) | (+10.0) | (+16.0) | (+14.0) | (+0.1) | (-0.5) | (+0.6) |
| *Llama-2-13B* | | | | | | | | | |
| CE | 26.3 | 26.9 | 28.2 | 14.0 | 14.0 | 14.0 | 45.4 | 50.4 | 52.3 |
| LongCE (Ours) | **26.4** | **28.5** | **28.9** | **20.0** | **18.0** | **18.0** | **55.1** | **61.9** | **62.5** |
| **Gain** | (+0.1) | (+1.6) | (+0.7) | (+6.0) | (+4.0) | (+4.0) | (+9.7) | (+11.5) | (+10.2) |

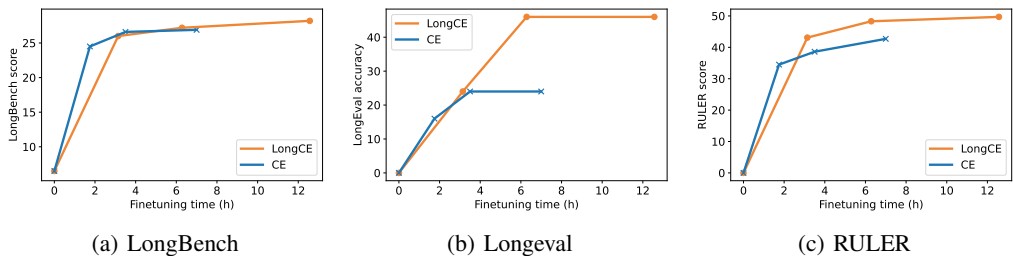

| (a) LongBench | (b) Longeval | (c) RULER |
|---|---|---|

Figure 7: Long-context fine-tuning performance (PG-19 dataset with EABF) vs. wall clock training time. LongCE demonstrates a stronger potential for enhancing long-context capabilities.

We visualize in Figure 7 how the long-context performance of models fine-tuned with LongCE and CE changes over the course of training time. Most of the time, fine-tuning with LongCE loss is a more efficient method. Additionally, in Appendix B.2, we find that by changing the hyperparameters of LongCE, *i.e.,* the short context-length $K$ and the sliding window length $d$, this overhead can be further reduced to 36%, with almost no loss in model performance.

## 5 CONCLUSION

In this paper, we offer a comprehensive explanation for why perplexity fails to reflect the long-context capabilities of LLMs. We find that as perplexity treats all tokens equally, it lacks sufficient attention on the key tokens that are crucial for long-context understanding. To address this, we propose a novel metric, LongPPL, which focuses on the key tokens in natural texts through a long-short context constrastive method. We empirically demonstrate the strong correlation with the long-context capabilities of LLMs as indicated by LongPPL and the performance on long-context benchmarks. In addition, we utilize the concept of LongPPL to propose the LongCE loss, which reweights the CE loss used in the long-context fine-tuning. By up-weighting the key tokens, LongCE leads to consistent improvements across multiple long-context benchmarks with up to 22% gains in LongEval accuracy. We hope our analysis and approaches can provide insights for a better understanding into the essence of long-context generation.

## ACKNOWLEDGEMENT

Yisen Wang was supported by National Key R&D Program of China (2022ZD0160300), National Natural Science Foundation of China (92370129, 62376010), and Beijing Nova Program (20230484344, 20240484642). Yifei Wang and Stefanie Jegelka were supported in part by the NSF AI Institute TILOS, and an Alexander von Humboldt Professorship.

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

# A DETAILED SETTINGS IN EXPERIMENTS

## A.1 IMPLEMENTATION DETAILS OF LONGPPL

**Sliding window algorithm to improve efficiency.** Since the calculation of LongPPL requires computing the LSD for each token $x_i, i \in [n]$, it necessitates calculating the probability under short context $P_\theta(x_i|\boldsymbol{s}_i)$ for $n - K$ times, where $K$ is the length of $\boldsymbol{s}_i$. Theoretically, the computational complexity of this process is $O((n - K)K^2)$. Since $K^2$ is typically larger than $n$ (e.g., when $K = 4096$, $K^2 = 16M$, which is much greater than $n = 32k$), this complexity far exceeds the normal $O(n^2)$ complexity of a standard long-context forward pass. As a result, the time cost of this process is quite significant.

To make this process more efficient, we use a sliding window algorithm to improve efficiency. Specifically, we introduce a step size $d$, which is smaller than the truncation length $l$ (we set it to $d = 1024$). When calculating the short-context probabilities of $x_i$ to $x_{i+d-1}$, we set the starting token of the context uniformly as $x_{i-l}$. Formally speaking, we have

$$s_{kd+i'} = (x_{(k-1)d}, ...x_{i'-1}),\tag{8}$$

where $k \in \mathbb{N}, 0 \leq i' < d$. This approach allows for the calculation of the short-context probabilities of $d$ tokens in a single forward pass, resulting in a complexity of $O((N - K)K^2/d)$. To access a better understanding on the selection of $K$ and $d$, please refer to Appendix B.2.

**Token matching method.** Since the used tokenizers between evaluator model $P_{\theta_0}$ and evaluated models $P_\theta$ could be different, we attempt to align the key tokens between different models. Formally, we define the *encoding* and *decoding* functions of tokenizers used in language models as $encode_P$ and $decode_P$. Let $\boldsymbol{t} = (t_1, ..., t_N)$ be the original text contains of $N$ characters, and $\boldsymbol{x} = (x_1, ..., x_n) = encode_{P_{\theta_0}}(\boldsymbol{t})$, $\boldsymbol{x}' = (x'_1, ..., x'_{n'}) = encode_{P_\theta}(\boldsymbol{t})$ be the token sequence encoded by $P_{\theta_0}$ and $P_\theta$, respectively. Let $\mathcal{X} = \{x_{k_i}\}_{i=1}^{n_k}$ be the set of key tokens calculated by the evaluator model $P_{\theta_0}$. We map these tokens to the text space as $\mathcal{T} = decode_{P_{\theta_0}}(\mathcal{X})$. Then, the key token set $\mathcal{X}'$ of the evaluated model is the maximal subset of $\boldsymbol{x}'$ which satisfies

$$decode_{P_\theta}(\mathcal{X}') \subseteq \mathcal{T}.\tag{9}$$

Besides, in Table 1, we also implement the LongPPL with the soft influence function $I_{\text{soft}}$ (Eq. (7)). In this approach, we implement an reweighting algorithm to transfer the weight between different tokenizers. Specifically, denote $\boldsymbol{w} = (w_1, ..., w_n)$ as the LSD weight on $\boldsymbol{x}$ calculated by $P_{\theta_0}$. The weight of $x'_i$ is defined as

$$w'_i = \sum_{t_j \in decode_{P_\theta}(x'_i)} w(t_j)/|decode_{P_\theta}(x'_i)|,\tag{10}$$

where $w(t_j)$ is the weight of the token that $t_j$ belongs to. This assigns the weight of $\boldsymbol{x}'$ with the string-level average of the weight in $\boldsymbol{x}$.

## A.2 IMPLEMENTATION DETAILS OF LONGCE

**Fine-tuning strategies.** For EABF (Zhang et al., 2024c), we adopt the identical settings in the original paper, with a RoPE base of 500k. For PI (Chen et al., 2023), we set the scaling factor to 8 since we want to extend the context window from 4k to 32k.

**Training details.** We use a learning rate of $2 \times 10^{-5}$ for Llama and $1 \times 10^{-6}$ for Mistral, with no weight decay and a linear warmup of 20 steps along with AdamW (Loshchilov, 2017) with $\beta_1 = 0.9$ and $\beta_2 = 0.95$. We apply a global batch of 64 on PG-19 and 8 on Pile-arxiv. We disable the sliding window mechanism when fine-tuning Mistral-7B-v0.1. We perform the experiments with 8 Nvidia A100 80GB GPUs using Pytorch (Paszke et al., 2019).

# B SUPPLEMENTARY EXPERIMENT RESULTS

## B.1 DETAILED RESULTS OF LONGPPL

We present the LongPPL calculated by different models in Table 5, and provide further visualization results for Mistral Large 2 in Figure 8.

Table 5: The perplexity-based metrics of various LLMs.

| Metric | | LongPPL | | PPL |
|---|---|---|---|---|
| Evaluator model | Qwen2-72B-Instruct | Mistral Large 2 | Llama-3.1-8B | - |
| Mixtral-8x7B-32k | 2.08 | 2.50 | 1.74 | 3.67 |
| FILM-7B-32k | 2.49 | 3.17 | 2.03 | 4.47 |
| Mistral-7B-32k | 2.68 | 3.49 | 2.19 | 4.25 |
| Qwen1.5-14B-128k | 2.97 | 2.93 | 2.33 | 5.23 |
| Qwen2-7B-128k | 2.99 | 2.73 | 2.29 | 4.97 |
| Phi-3-small-128k | 2.98 | 2.86 | 2.41 | 5.42 |
| CLEX-7B-64k | 3.70 | 4.60 | 2.92 | 4.13 |
| Yi-6B-200k | 3.62 | 3.92 | 2.86 | 5.11 |
| Yarn-7B-128k | 3.67 | 4.88 | 3.10 | 4.17 |

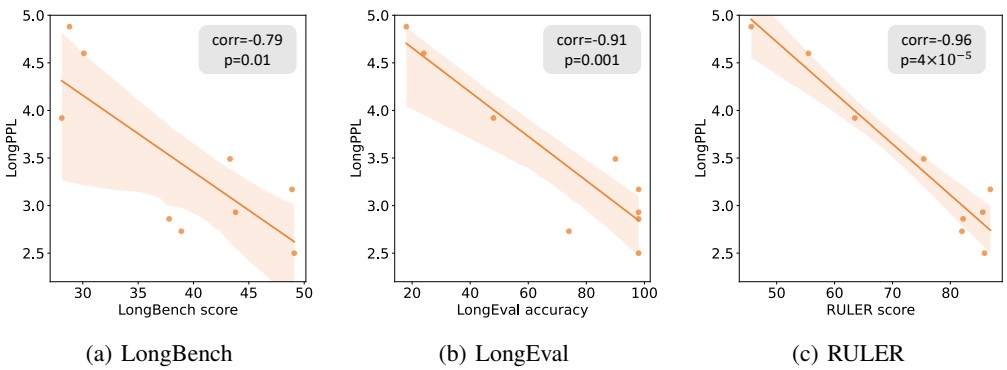

| (a) LongBench | (b) LongEval | (c) RULER |
|---|---|---|

Figure 8: Correlation between LongPPL on GovReport and long-context benchmarks. LongPPL is calculated using Mistral Large 2.

## B.2 ABLATION STUDY

**LCL.** In the calculation of LongPPL, we employ LCL as an assistant for our core criterion, LSD, in selecting key tokens. In Figure 9, we demonstrate the LongPPL calculated without the LCL criterion. This version of LongPPL hardly has correlation with the long-context benchmark, showing that LCL is an indispensable part for LongPPL.

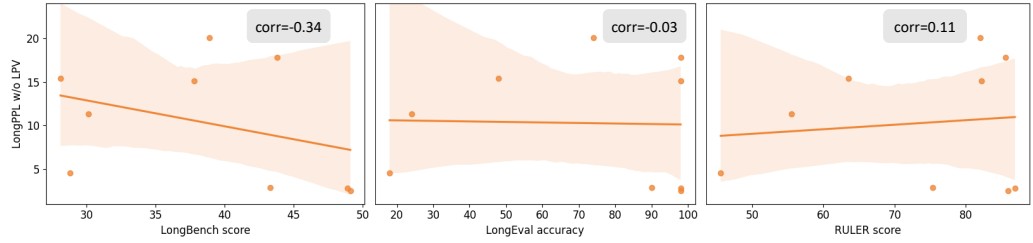

Figure 9: LongPPL without LCL.

**Evaluator model.** In the main text, we use a evaluator model $\theta_0$ to identify the key tokens. To validate the necessity of this approach, we calculate LongPPL using the model itself as the evaluator, as shown in Table 6. The results indicate that most models achieve similar LongPPL scores, suggesting that this self-evaluated version of LongPPL does not reflect the models' long-context capabilities.

**Hyperparameters of LongCE.** In the computation of LongCE, several hyperparameters are utilized, including the short context window length $K$ and sliding window length $d$ used in calculating LSD. Here, we design ablation experiments to analyze the selection of these hyperparameters, as

Table 6: LongPPL using the evaluated model itself to calculate the key tokens.

|  | Mixtral | FILM | Mistral | Qwen1.5 | Qwen2 | Phi-3 | CLEX | Yi | Yarn |
|---|---|---|---|---|---|---|---|---|---|
| LongPPL | 1.67 | 1.64 | 1.68 | 1.67 | 1.65 | 1.65 | 1.68 | 1.75 | 1.92 |

Table 7: The performance and time cost of LongCE on long-context benchmarks under different hyperparameter settings of $K$ and $d$. For the time cost, we report the wall-clock time for training 200 steps.

| Training steps | Total training time / h 200 | LongBench 50 | 100 | 200 | LongEval 50 | 100 | 200 | RULER 50 | 100 | 200 |
|---|---|---|---|---|---|---|---|---|---|---|
| | *Setting A (PG-19 dataset with EABF)* | | | | | | | | | |
| CE | 7.0 | 24.5 | 26.6 | 26.9 | 16.0 | 24.0 | 24.0 | 34.5 | 38.6 | 42.7 |
| LongCE ($K$ = 4k, $d$ = 1k, default) | 12.5 (+79%) | **26.0** | **27.2** | **28.2** | 24.0 | 46.0 | 46.0 | 43.1 | 48.3 | 49.7 |
| LongCE ($K$ = 1k, $d$ = 1k) | 10.0 (+43%) | 25.3 | 25.8 | 26.9 | 20.0 | 48.0 | 48.0 | **45.6** | **51.1** | **55.9** |
| LongCE ($K$ = 4k, $d$ = 4k) | 9.5 (+36%) | 25.4 | 25.8 | 25.8 | **28.0** | **56.0** | 56.0 | 42.5 | 48.0 | 51.2 |
| LongCE ($K$ = 4k, $d$ = 512) | 17.5 (+150%) | 25.4 | 25.8 | 27.3 | 26.0 | 48.0 | **60.0** | 42.4 | 50.1 | 54.4 |

shown in Table 7. The results reveal that, on one hand, increasing $K$ or decreasing $d$ significantly improves the efficiency of LongCE (from +79% to +36%/+43%). On the other hand, under these settings, although the model's performance on real-world tasks (LongBench) slightly decreases, it achieves substantial improvements on synthetic tasks (LongEval, RULER). This suggests that LongCE still holds potential for further efficiency enhancements.

## B.3 FINE-GRAINED RESULTS OF LONGCE

In this section, we provide more detailed LongBench scores of the models from the experiments in section 4.2, as shown in Table 8. We observe that the models finetuned by LongCE outperforms the model finetuned with CE primarily in single/multi-document QA, summarization and synthetic tasks (including retrieval and counting tasks). This also explains why LongCE can significantly outperform CE on LongEval and RULER, as their synthetic tasks primarily assess models' retrieval, summarization, and QA capabilities in long-context scenarios.

Table 8: Detailed scores of LongBench in Table 3.

| Task Domains | Single-Document QA | Multi-Document QA | Summarization | Few-shot Learning | Code Completion | Synthetic Tasks | Avg. |
|---|---|---|---|---|---|---|---|
| | *Setting A (PG-19 dataset with EABF)* | | | | | | |
| CE (50 steps) | 4.4 | 1.1 | 15.5 | 66.7 | 59.7 | 0.0 | 24.5 |
| CE (100 steps) | 5.9 | 2.0 | 21.9 | **67.5** | 61.8 | 0.4 | 26.6 |
| CE (200 steps) | 6.9 | 2.3 | 22.8 | 66.8 | **61.9** | 0.4 | 26.9 |
| LongCE (50 steps) | 7.6 | 2.1 | 22.0 | 66.1 | 57.9 | 0.5 | 26.0 |
| LongCE (100 steps) | 7.7 | 3.3 | 22.5 | 65.7 | 61.6 | 2.3 | 27.2 |
| LongCE (200 steps) | **9.3** | **4.8** | **23.9** | 66.0 | **61.9** | **3.2** | **28.2** |
| | *Setting B (PG-19 dataset with PI)* | | | | | | |
| CE (50 steps) | 3.1 | 3.2 | 12.9 | 65.3 | 59.8 | 1.6 | 24.3 |
| CE (100 steps) | 4.1 | 3.5 | 17.5 | 65.2 | 59.9 | 1.8 | 25.3 |
| CE (200 steps) | 5.6 | 4.0 | 15.4 | **66.0** | **60.3** | 1.0 | 25.4 |
| LongCE (50 steps) | 4.5 | 2.2 | 15.6 | 63.1 | 58.4 | 2.7 | 24.4 |
| LongCE (100 steps) | 4.6 | 1.7 | 17.7 | 64.1 | 59.0 | **2.8** | 25.0 |
| LongCE (200 steps) | **6.0** | **4.3** | **19.0** | 63.6 | 59.2 | 2.7 | **25.8** |
| | *Setting C (Pile-arxiv dataset with EABF)* | | | | | | |
| CE (50 steps) | 1.7 | 0.0 | 0.0 | 50.2 | 38.2 | 0.0 | 15.0 |
| CE (100 steps) | 4.2 | 5.4 | 4.9 | **65.0** | 58.9 | 0.0 | 23.1 |
| CE (200 steps) | **5.1** | **7.1** | 7.6 | 64.3 | 58.7 | 0.0 | 23.8 |
| LongCE (50 steps) | 3.5 | 0.0 | 2.6 | 52.9 | 46.7 | 0.0 | 17.6 |
| LongCE (100 steps) | 4.2 | 5.3 | 10.0 | 64.3 | 59.1 | 1.0 | 24.0 |
| LongCE (200 steps) | 3.7 | 6.1 | **14.3** | 64.7 | **59.8** | **1.3** | **25.0** |

Table 9: Detailed results of experiments in Figure 2, including the accuracy on LongEval, and perplexity tested on answer and non-answers tokens, respectively.

| Prompt Length | 2k | 3k | 4k | 5k | 7k | 9k | 11k | 13k | 15k | 17k | 19k | 21k | 23k | 25k | 28k |
|---|---|---|---|---|---|---|---|---|---|---|---|---|---|---|---|
| | | | | | | *Yi-6B-200K* | | | | | | | | | |
| LongEval accuracy / % | 100.0 | 94.0 | 84.0 | 76.0 | 76.0 | 64.0 | 68.0 | 54.0 | 60.0 | 58.0 | 46.0 | 44.0 | 50.0 | 52.0 | 48.0 |
| PPL (answer tokens) | 1.49 | 1.47 | 1.59 | 1.64 | 1.91 | 2.00 | 1.98 | 2.29 | 2.28 | 2.15 | 2.39 | 2.11 | 2.23 | 2.32 | 2.08 |
| PPL (non-answer tokens) | 2.15 | 2.17 | 2.12 | 2.18 | 2.18 | 2.20 | 2.27 | 2.25 | 2.25 | 2.23 | 2.23 | 2.21 | 2.22 | 2.25 | 2.24 |
| | | | | | | *CLEX-7B-64K* | | | | | | | | | |
| LongEval accuracy / % | 82.0 | 34.0 | 84.0 | 82.0 | 58.0 | 62.0 | 58.0 | 56.0 | 50.0 | 44.0 | 46.0 | 24.0 | 22.0 | 28.0 | 24.0 |
| PPL (answer tokens) | 1.31 | 2.33 | 1.23 | 1.33 | 1.47 | 1.43 | 1.51 | 1.54 | 1.63 | 1.78 | 1.89 | 2.23 | 2.50 | 2.61 | 2.59 |
| PPL (non-answer tokens) | 2.22 | 2.31 | 2.17 | 2.18 | 2.10 | 2.16 | 2.17 | 2.14 | 2.14 | 2.15 | 2.15 | 2.18 | 2.20 | 2.24 | 2.24 |

## B.4 DETAILED RESULTS OF THE EXPERIMENTS IN SECTION 2.1

In Table 9, we present the detailed results from the experiments in Figure 2(b) and 2(c).

## B.5 NEEDLE-IN-A-HAYSTACK RESULTS

In this section, we conduct the standard Needle-in-a-Haystack (NIAH) evaluation to evaluate models' long-context capability when context lengths is greater than 32K.

We first test the models obtained in the main text, which are fine-tuned on 32K-length texts. As shown in figure 10, LongCE achieves a score of 10 on 5 out of 6 questions at the 40K length and 2 out of 6 questions at the 48K length, outperforming CE, which achieves a score of 10 on 2 out of 6 and 0 out of 6 questions, respectively. Therefore, LongCE demonstrates a longer effective context length.

Additionally, to demonstrate the generalization ability of LongCE on longer context lengths, we extend the context window of both models by increasing their RoPE base from 500K to 2M. The corresponding NIAH results are shown in Figure 11. The results show that model finetuned with LongCE answers all questions correctly at the 64K length and achieves a score of 10 on 32 sequences with lengths of $\geq$32K, while CE only achieves this on 26 sequences. This indicates that LongCE can generalize well at longer lengths.

## B.6 LONGCE'S PERFORMANCE ON NON-LONG-CONTEXT LANGUAGE TASKS

In this section, we experimentally investigate whether LongCE will adversely impact non-long-context capabilities. In Table 10, we present the model performance on 6 common language tasks, *i.e.,* MMLU (Hendrycks et al., 2021), ARC-Challenge (Clark et al., 2018), RACE (Lai et al., 2017), BigBench Hard (Suzgun et al., 2023), TruthfulQA (Lin et al., 2022), and CommonsenseQA (Talmor et al., 2019). The results show that for non-long-context tasks, the performance of the model trained with LongCE is nearly identical to that of the model trained with CE, indicating that the long-context-specific characteristics of LongCE do not negatively affect the model's performance on tasks involving normal-length context compared to the baseline.

Table 10: The performance of models fine-tuned with CE and LongCE on non-long-context tasks. The models are finetuned with 200 steps under the setting A in Table 3.

| Models | MMLU | ARC-C | RACE | BBH | TruthfulQA | CommonsenseQA | Avg. |
|---|---|---|---|---|---|---|---|
| Llama-2-7B | 41.8 | 43.3 | 39.5 | 39.4 | 34.5 | 32.9 | 38.6 |
| +CE (baseline) | **40.8** | 42.8 | **40.3** | 36.4 | 29.3 | **31.5** | **36.9** |
| +LongCE (ours) | 39.9 | **43.9** | 39.3 | **37.5** | **30.0** | 30.8 | **36.9** |

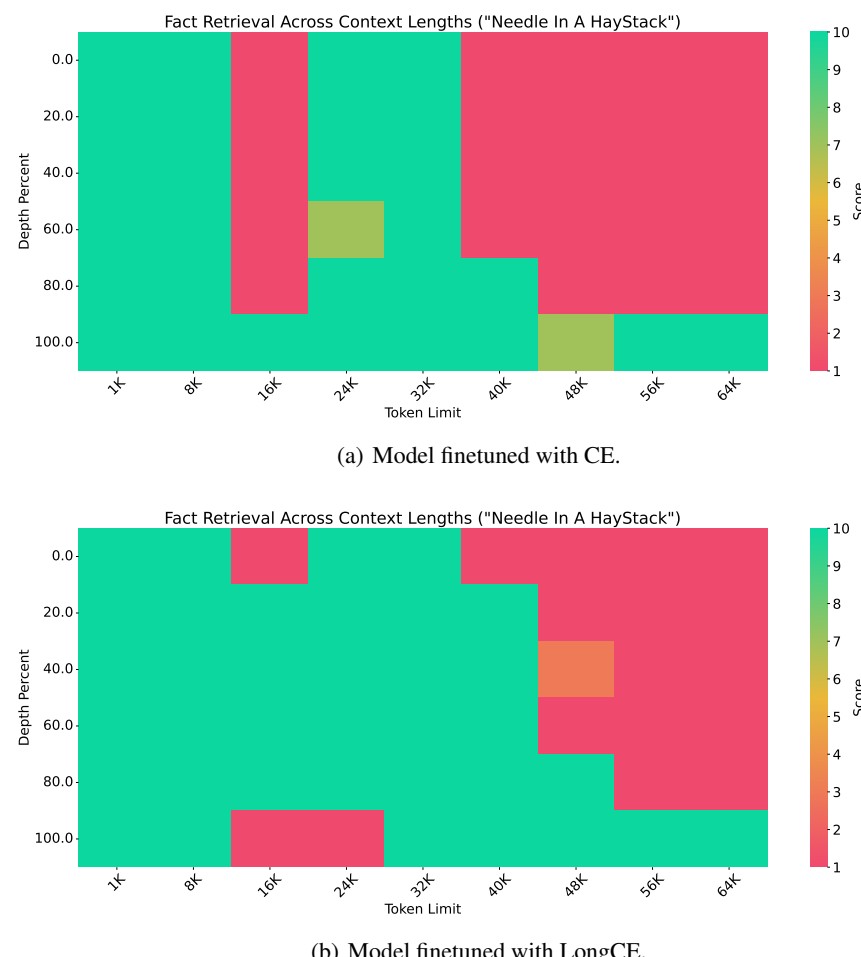

(a) Model finetuned with CE.

(b) Model finetuned with LongCE.

Figure 10: Needle-in-a-haystack results of models trained with PG-19 datasets & EABF for 200steps.

## B.7 SUBSTITUTING KEY TOKENS WITH RE-OCCURRED N-GRAM

In this section, we examine whether LongPPL works by retrieving the frequent N-gram in the context, as concerned in recent works (Sun et al., 2021; Arora et al., 2024). We calculate perplexity solely on the re-occurred N-gram (word-level, $N > 2$) in the inputs, and present the correlation coefficients with the benchmarks in Table 11.

Table 11: The correlation coefficients between PPL calculated on re-occurred N-gram, and the benchmarks.

|              | LongBench | LongEval | RULER |
|--------------|-----------|----------|-------|
| PPL          | -0.11     | 0.24     | 0.27  |
| PPL (N-gram) | -0.41     | -0.10    | -0.05 |
| LongPPL      | -0.96     | -0.86    | -0.84 |

The results show that PPL on re-occurred N-grams has much weaker correlation with model's long-context capabilities. This indicates that LongPPL's powerful ability to capture long-context-related information cannot be simply explained by N-grams.

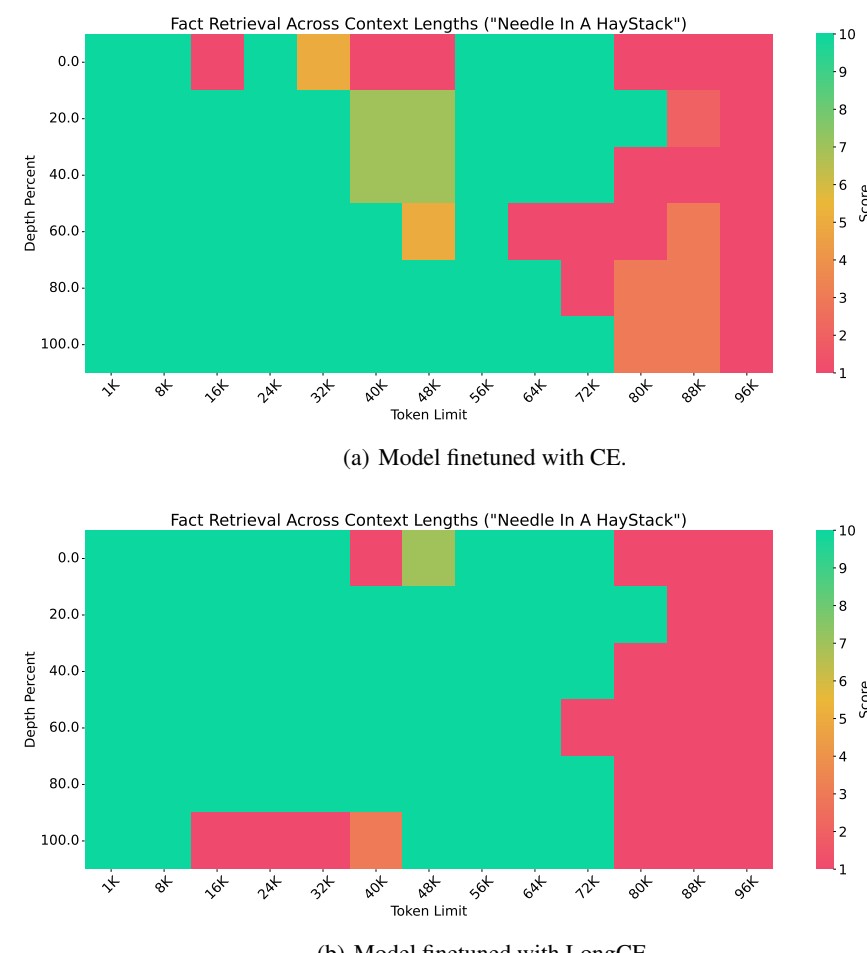

(a) Model finetuned with CE.

(b) Model finetuned with LongCE.

Figure 11: Needle-in-a-haystack results of models trained with PG-19 datasets & EABF for 200steps. We increase the RoPE base from 500k to 2M after finetuning.

## B.8 TIME CONSUMPTION OF LONGPPL

In Table 12, we test the time cost of LongPPL. It can be observed that the time cost of calculating LongPPL using the 8B model as the evaluator is approximately 3∼4 times that of calculating PPL, while the overhead for using the 72B model is much higher.

Although the computational overhead of LongPPL is non-negligible, we believe that such a computational cost will not have a substantial impact on the practicality of LongPPL. On the one hand, if users employ LongPPL as a benchmark, key tokens can be calculated offline, resulting in no online computation overhead. On the other hand, if LongPPL is used as an evaluation metric during training, its computational overhead is negligible compared to the overall training cost (as evaluation steps are typically sparse during training).

Table 12: The time consumption of LongPPL. The values in the table represent the average seconds required per sequence.

|  | PPL | LongPPL (Llama-3.1-8B) | LongPPL (Qwen2-72B-Instruct) |
|---|---|---|---|
| Mistral-7B | 2.8 | 11.3 (+8.5, +304%) | 56.4 (+53.6, +2014%) |
| Mixtral-8x7B (47B) | 4.2 | 13.5 (+9.3, +221%) | 58.4 (+54.2, +1390%) |

## C  RELATED WORK

**Long-context Modeling.** Due to practical demands, numerous recent works have emerged that aim to enable large models to handle long contexts through improvements in architecture or algorithms. One mainstream direction is the study of positional encodings with length extrapolation capabilities, including Alibi (Press et al., 2021), xPOS (Sun et al., 2023), Kerple (Chi et al., 2022), and various RoPE (Su et al., 2024) variants (Chen et al., 2023; Zhang et al., 2024c; Chen et al., 2024a; Xiong et al., 2024; Peng et al., 2024). Others pay more attention to architecture improvements, using sparse attention mechanisms to prevent models from attending to overly long sequences (Han et al., 2023; Xiao et al., 2024; Chen et al., 2024b; Ding et al., 2023), or exploring the use of recurrent mechanisms to compress and store key information from long texts, thereby effectively increasing the context window (Zhang et al., 2024a; Bulatov et al., 2023; Martins et al., 2022).

**Long-context Evaluation.** Recent studies have introduced several benchmarks to evaluate the long-context performance in downstream tasks. A widely used type of benchmark is retrieval-based synthetic task, including needle-in-a-haystack (Kamradt, 2023), passkey-retrieval (Mohtashami & Jaggi, 2023) and LongEval (Li et al., 2023a). Some evaluation suites have also been gradually introduced, such as LongBench (Bai et al., 2023b), RULER (Hsieh et al., 2024), ZeroSCROLLS (Shaham et al., 2023), including document question answering, summarization, few-shot learning, code completion, and other synthetic tasks, thereby offering a more thorough evaluation of a model's long-context abilities. To further enhance the context length of the evaluation data, InfiniteBench (Zhang et al., 2024b) has introduced evaluation data exceeding 100K tokens. In this paper, we analyze the correlation between the Perplexity metric and specific evaluation tasks and propose an alternative LongPPL metric, which can better align the model's long-context performance on downstream tasks.

**Re-weighting methods in language model training.** Re-weighting methods for language model training have been extensively studied, with a focus on enhancing model performance (Lin et al., 2024), improving training efficiency (Clark et al., 2022), and addressing token imbalance (Luo et al., 2023; Hu et al., 2023; Gu et al., 2020; Wang et al., 2020). Many works have also explored re-weighting through data selection techniques, addressing a wide range of challenges such as data quality (Li et al., 2023b), data diversity (Liu et al., 2023), and distribution matching (Li et al., 2023c; Ni et al., 2024). However, few of these works focus on re-weighting tokens to enhance a model's long-context performance. The most recent and closely related work to ours is LongRecipe (Hu et al., 2024b), which re-weights tokens based on distribution shifts in model predictions during training. This approach does not capture the essential characteristics of key tokens. In contrast, our method directly re-weights tokens according to their dependence on long-context information, providing a more fundamental and targeted solution.

# D  MODELS

The models used in this paper are shown in Table 13.

Table 13: Information of the models used in this paper.

| Model | Size | Context Length | Huggingface |
|---|---|---|---|
| Llama-2-7B (Touvron et al., 2023) | 7B | 4K | meta-llama/Llama-2-7b-hf |
| Llama-2-13B (Touvron et al., 2023) | 13B | 4K | meta-llama/Llama-2-13b-hf |
| Llama-3.1-8B (Dubey et al., 2024) | 8B | 128K | meta-llama/Llama-3.1-8B |
| Mixtral (Jiang et al., 2024) | 8x7B | 32K | mistralai/Mixtral-8x7B-Instruct-v0.1 |
| Mistral-v0.1 (Jiang et al., 2023) | 7B | 8K | mistralai/Mistral-7B-v0.1 |
| Mistral (Jiang et al., 2023) | 7B | 32K | mistralai/Mistral-7B-Instruct-v0.2 |
| Mistral Large 2 (Jiang et al., 2023) | 123B | 128K | mistralai/Mistral-Large-Instruct-2407 |
| Qwen1.5 (Bai et al., 2023a) | 14B | 128K | Qwen/Qwen1.5-14B |
| Qwen2-7B (Yang et al., 2024) | 7B | 128K | Qwen/Qwen2-7B |
| Qwen2-72B (Yang et al., 2024) | 72B | 128K | Qwen/Qwen2-72B-Instruct |
| FILM (An et al., 2024) | 7B | 32K | In2Training/FILM-7B |
| Phi-3 (Abdin et al., 2024) | 7B | 128K | microsoft/Phi-3-small-128k-instruct |
| CLEX (Chen et al., 2024a) | 7B | 64k | DAMO-NLP-SG/CLEX-LLaMA-2-7B-64K |
| Yi (Young et al., 2024) | 6B | 200K | 01-ai/Yi-6B-200K |
| Yarn (Peng et al., 2024) | 7B | 128K | NousResearch/Yarn-Mistral-7b-128k |

## E  DEMONSTRATION FOR THE SELECTED KEY TOKENS

---

**Demonstration for the selected key tokens in GovReport**

............

Even though it has reimposed all U.S. sanctions on Iran, the Trump Administration has issued some exceptions that are provided for under the various U.S. sanctions laws, including the following:  As noted above, on November 5, 2018, eight countries were given the SRE to enable them to continue transactions with Iran's Central Bank and to purchase Iranian oil.  At an April 10 hearing of the Senate Foreign Relations Committee, Secretary Pompeo appeared to indicate that the SREs would be renewed.  However, on April 22 the Administration announced termination of the SREs as of their expiration on May 2, 2019.  On May 3, the Administration ended some waivers under IFCA and various antiproliferation laws (discussed above) that allow international technical assistance to Iran's three nuclear sites permitted to operate under the JCPOA—the Fordow facility, the Bushehr nuclear power reactor, and the Arak heavy water plant.  The Administration ended the waiver that enabled Rosatom (Russia) to remove Iran's LEU that exceeds the 300kg allowed stockpile, and that allowed Iran to export heavy water that exceeded the limits on that product to Oman.  The waiver limitations also will prohibit the expansion of the Bushehr reactor by any supplier. In response, President Rouhani announced that Iran would no longer abide by the JCPOA stockpile limits. The Administration waived Section 1247(e) of IFCA to enable Iraq to continue paying for purchases of natural gas from Iran.  The waiver term for that section is up to 180 days, but the Administration has been providing the waiver for 90-day increments. The Administration has issued the permitted IFCA exception for Afghan reconstruction to enable India to continue work at Iran's Chahbahar Port. A U.S. State Department official told Afghan leaders in mid-May 2019 that the exception would continue.  The Administration has renewed the licenses of certain firms to enable them to continue developing the Rhum gas field in the North Sea that Iran partly owns.

............

The JCPOA did not commit the United States to suspend U.S. sanctions on Iran for terrorism or human rights abuses, on foreign arms sales to Iran or sales of proliferation-sensitive technology such as ballistic missile technology, or on U.S.-Iran direct trade (with the selected exceptions of the latter discussed above).  The sanctions below remained in place during JCPOA implementation and remain in effect now: E.O. 12959, the ban on U.S. trade with and investment in Iran; E.O. 13224 sanctioning terrorism entities, any sanctions related to Iran's designation as a state sponsor or terrorism, and any other terrorism-related sanctions.  The JCPOA does not commit the United States to revoke Iran's placement on the terrorism list; E.O. 13382 sanctioning entities for proliferation; the Iran-Iraq Arms Non-Proliferation Act; the Iran-North Korea-Syria Non-Proliferation Act (INKSNA); the section of ISA that sanctions WMD- and arms-related transactions with Iran; E.O. 13438 on Iran's interference in Iraq and E.O. 13572 on repression in Syria; Executive Orders (E.O. 13606 and E.O. 13628) and the provisions of CISADA, ITRSHRA, and IFCA that pertain to human rights or democratic change in Iran; all sanctions on the IRGC, military, proliferation-related, and human rights- and terrorism-related entities, which were not "delisted" from sanctions; Treasury Department regulations barring Iran from access to the U.S. financial system.  Foreign banks can pay Iran in dollars out of their existing dollar supply, and the Treasury Department revised its guidance in October 2016 to stress that such transactions are permitted.

............

---

