# OpenReview forum: "What is Wrong with Perplexity for Long-context Language Modeling?"
_ICLR.cc/2025/Conference — ICLR 2025 Poster_

### Official Review · Reviewer_KQKC · 2024-10-24

**Soundness:** 4
**Presentation:** 4
**Contribution:** 4
**Rating:** 8
**Confidence:** 5

**Summary:**

This paper first tries to understand why perplexity metric fail to reflect long-context ability of LLMs. For instance it has been shown that perplexity of LLMs shows almost no correlation with their long-context performance measured on a dataset such as Longbench.

Authors show the reason is because perplexity is computed over all tokens of the answer to a query, wether this tokens are ‘key’ or not to answer to the query. Consequently, perplexity computed equally over all tokens do not represent long-context performance. Therefore,  computing ppl by averaging only on key tokens (instead of all tokens)  avoid this dilution and correlates better with long context benchmarks scores such as the ones measured on Longbench.

A longPPL measure that takes into account this observation is proposed by the authors, where they compute perplexity by averaging solely on the selected key tokens.
But how do the authors extract the ‘key’ tokens from the natural text in the answer to a query ?
 This is done by introducing a simple and elegant metric which consists in computing the difference - for each token x - of the probability p(x/long_context) and p(x/shortened_context). This metric called LPG score, measures the improvement of prediction accuracy induced only by the long-context => a high value indicates that long context plays an important role in the prediction of x, so x is a key token and should be taken into account to compute LongPPL (and it should not be taken into account - for compute the PPL - otherwise).

Building on this finding, they develop an efficient long-context training strategy by prioritizing key tokens. Specifically, they introduce the LongCE (Long-context Cross-Entropy) loss, which assigns higher weights to key tokens, identified dynamically by the model itself.
Experiments across various LLMs demonstrate that fine tuning a model using the LongCE  loss consistently outperforms traditional CE loss, across multiple LLMs and multiple long context benchmarks.

**Strengths:**

-Paper very clearly presented and very progressive, starting from identifying a problem, explain why it occurs and propose an efficient solution for it

- convincing explanation on why PPL is not a good indicator of quality for long context LLMs (with illustrations on a long context benchmark)

-Fine tuning with longCE loss leads to improvements on several long context benchmarks and for several LLMs ; very convincing results

**Weaknesses:**

-I don’t fully agree this the claim « the evaluation of long-context capabilities still heavily relies on perplexity (PPL) as the de facto metric » as we see now most papers relying on long-context benchmarks or use the needle-in-the-haystack test to evaluate long-context LLMs

-The reference to ppl is not the good one (Bengio did not invent perplexity!), you should credit people who actually were the first to introduce this metric. Perplexity was originally introduced in 1977 in the context of speech recognition by Frederick Jelinek, Robert Leroy Mercer, Lalit R. Bahl, and James K. Baker. See https://en.wikipedia.org/wiki/Perplexity

-I’d like to learn more about the computational implications of calculating LongPPL and LongCE, especially since the LPG score needs to be computed for each token (see my question below). I'm particularly curious about the costs associated with LongCE, as described in Section 3.2. While the Implementation details of LongPPL in the Appendix were helpful, I still want to understand more about how computationally expensive this process is.

**Questions:**

-computed longPPL and longCE (as opposed to PPL) must be more costly; how much more costly (especially a you need to compute the LPG score for each token at training time) ?

-do you fine tune with longCE only or with a combination of CE and longCE ?

-what about using the needle-in-the-haystack test for your evaluation of long context models ?

---

> ### Author Response · Authors · 2024-11-20
> **Response to Reviewer KQKC (Part 1/2)**
>
> We appreciate Reviewer KQKC for giving highly positive feedback on the solidness of our paper and the persuasiveness of our experimental results. Now we address your concerns in the following points.
>
> ---
>
> **Q1.** I don’t fully agree this the claim « the evaluation of long-context capabilities still heavily relies on perplexity (PPL) as the de facto metric » as we see now most papers relying on long-context benchmarks or use the needle-in-the-haystack test to evaluate long-context LLMs.
>
> **A1.** Thanks for pointing out the inaccuracies in the expression in our paper. We have modified it to “the evaluation of long-context capabilities still **widely uses** perplexity (ppl) as the de facto metric.”
>
> ---
>
> **Q2.** The reference to ppl is not the good one (Bengio did not invent perplexity!), you should credit people who actually were the first to introduce this metric. Perplexity was originally introduced in 1977 in the context of speech recognition.
>
> **A2.** Thanks for pointing to the original reference. Following your suggestion, we have updated the paper with the correct citations.
>
> ---
>
> **Q3.** I’d like to learn more about the computational implications of calculating LongPPL and LongCE, especially since the LPG score needs to be computed for each token (see my question below). I'm particularly curious about the costs associated with LongCE, as described in Section 3.2. While the Implementation details of LongPPL in the Appendix were helpful, I still want to understand more about how computationally expensive this process is.
>
> **A3.** The calculation of both LongPPL and LongCE involves two steps: (1) One forward pass on the full text, and (2) several forward passes on short texts (to obtain the LPG score for each token). Step 1, which involves calculations for both PPL (CE) and LongPPL (LongCE), has a time complexity of  $O(N^2)$ , where N is the input sequence length (N=32k in this paper). Step 2 represents the computational overhead of LongPPL (LongCE), with a time complexity of  $O((N-K)K^2/d)$, where K is the length of the short context(K=4k in this paper), and d is the sliding window length (d=1k in this paper). Under the parameter settings in this paper, the computational overhead for LongCE is approximately 80% of the cost of CE.
>
> Specifically, if we calculate the LPG for each token according to its original definition in Eq. 5, then a total of  N-K  forward passes of length K are needed, resulting in a computational complexity of  $O((N-K)K^2)$. To make this process more efficient, we utilize the sliding window algorithm described in Appendix A.1. When the sliding window is set to d , we can obtain the LPG values for **d** tokens in **a single forward pass** of length  K + d , whereas previously, we could only obtain the LPG value for **one** token in a forward pass of length K. Under this algorithm, step 2 only requires approximately $\frac{N-K}{d}$ forward passes on a sequence, reducing the complexity of the algorithm to $O((N-K)K^2/d)$.
>
> Additionally, in terms of memory usage, LongPPL incurs extra overhead from loading the evaluator model. However, since LongCE uses the same model for both evaluation and training, it only introduces intermediate storage overhead from the inference of short context.
>
> We have included the above discussion in Appendix A.1.
>
> ---
>
> **Q4.** Computing longPPL and longCE (as opposed to PPL) must be more costly; how much more costly (especially a you need to compute the LPG score for each token at training time) ?
>
> **A4.** In A3, we have already elaborated on the theoretical computational complexity of LongPPL (LongCE). Next, we will further explain their specific time overhead.
>
> For LongPPL, we have added the following table detailing the time costs.
>
> | Evaluated model \ Time cost (second / seq) | PPL | LongPPL (Llama-3.1-8B) | LongPPL (Qwen2-72B) |
> | --- | --- | --- | --- |
> | Mistral-7B | 2.8 | 11.3 (+8.5, +304%) | 56.4 (+53.6, +2014%) |
> | Mixtral-8x7B (47B) | 4.2 | 13.5 (+9.3, +221%) | 58.4 (+54.2, +1390%) |
>
> It can be observed that the time cost of calculating LongPPL using the 8B model as the evaluator is approximately 3~4 times that of calculating PPL, while the overhead for using the 72B model is much higher.
>
> Although the computational overhead of LongPPL is non-negligible, we believe that such a computational cost will not have a substantial impact on the practicality of LongPPL. On the one hand, if users employ LongPPL as a benchmark, key tokens can be calculated offline, resulting in no online computation overhead. On the other hand, if LongPPL is used as an evaluation metric during training, its computational overhead is negligible compared to the overall training cost (as evaluation steps are typically sparse during training). We have included the above results in Appendix B.7.

---

> ### Author Response · Authors · 2024-11-20
> **Response to Reviewer KQKC (Part 2/2)**
>
> For LongCE, we conducted an additional set of ablation experiments. In the following table, we present the long-context performance of models finetuned with LongCE under different values of K and d, which directly influence its computational efficiency.
>
> | PG-19 dataset with EABF | Total Training time (200 step) / h | LongBench |  |  | LongEval |  |  | RULER |  |  |
> | --- | --- | --- | --- | --- | --- | --- | --- | --- | --- | --- |
> |  |  | 50 steps | 100 steps | 200 steps | 50 steps | 100 steps | 200 steps | 50 steps | 100 steps | 200 steps |
> | K=4096, d=512 | 17.5 (+150%) | 25.4 | 25.8 | 27.3 | 26.0 | 48.0 | **60.0** | 42.4 | 50.1 | 54.4 |
> | K=4096, d=1024 (default) | 12.5 (+79%) | **26.0** | **27.2** | **28.2** | 24.0 | 46.0 | 46.0 | 43.1 | 48.3 | 49.7 |
> | K=4096, d=4096 | 9.5 (+36%) | 25.4 | 25.8 | 25.8 | **28.0** | **56.0** | 56.0 | 42.5 | 48.0 | 51.2 |
> | K=1024, d=1024 | 10.0 (+43%) | 25.3 | 25.8 | 26.9 | 20.0 | 48.0 | 48.0 | **45.6** | **51.1** | **55.9** |
> | CE | 7 | 24.5 | 26.6 | 26.9 | 16.0 | 24.0 | 24.0 | 34.5 | 38.6 | 42.7 |
>
> First, it can be observed that increasing K or decreasing d significantly improves the efficiency of LongCE (from +79% to +36%/+43%). Second, under these settings, although the model’s performance on real-world tasks (LongBench) slightly decreases, it achieves substantial improvements on synthetic tasks (LongEval, RULER). This suggests that LongCE still holds potential for further efficiency enhancements.
>
> We have included the above result in Appendix B.2.
>
> ---
>
> **Q5.** Do you fine tune with longCE only or with a combination of CE and longCE ?
>
> **A5.** We fine-tune with LongCE only. To maintain a balance between general language modeling ability and long-context capability, one could also consider combine CE and LongCE during training.
>
> ---
>
> **Q6.** What about using the needle-in-the-haystack test for your evaluation of long context models ?
>
> **A6.** To evaluate the long-context capabilities over 32k, we conduct the standard Needle-in-a-Haystack (NIAH) evaluation in Appendix B.5. We first test the models obtained in our main text, which are fine-tuned on 32K-length texts. As shown in figure 11, LongCE achieves a score of 10 on 5 out of 6 questions at the 40K length and 2 out of 6 questions at the 48K length, outperforming CE, which achieves a score of 10 on 2 out of 6 and 0 out of 6 questions, respectively. Therefore, LongCE demonstrates a longer effective context length.
>
> Additionally, to demonstrate the generalization ability of LongCE on longer context lengths, we extend the context window of both models by increasing their RoPE base from 500K to 2M. The corresponding NIAH results are shown in Figure 12. The results show that model finetuned with LongCE answers all questions correctly at the 64K length and achieves a score of 10 on 32 sequences with lengths of ≥32K, while CE only achieves this on 26 sequences. This indicates that LongCE can generalize well at longer lengths.
>
> ---
>
> Thanks again for your detailed comments. Please let us know if there is more to clarify.

---

> > ### Comment · Reviewer_KQKC · 2024-11-22
> > **acknowledging the authors' answers**
> >
> > hi, I acknowledging the authors' answers.
> > I am happy especially with the discussion related to computation complexity from both theoretical and practical point of view and putting this in the Appendix will definitely make the paper stronger.
> > Happy as well with the needle-in-a-haystack test added in Appendix B.5
> > All this confirms my rating (8: accept, good paper).

---

> > > ### Author Response · Authors · 2024-11-22
> > >
> > > Thanks once again for your encouraging feedback, and we are glad to see that our responses resolved your concerns. Have a great day!

---

### Official Review · Reviewer_At6Q · 2024-10-28

**Soundness:** 2
**Presentation:** 3
**Contribution:** 3
**Rating:** 6
**Confidence:** 4

**Summary:**

This paper proposes a new metric LongPPL and a new training objective LongCE. The LongPPL is calculated over selected tokens that satisfy the following conditions when evaluated with an evaluator model: (a) the tokens have higher probability conditioned on long context than short context (b) the tokens are predicted with high probability. Results show that the proposed metric better correlates with long-context downstream tasks than standard perplexity. The proposed LongCE loss adapts the standard focal loss by reweighting the tokens based on their performance given long and short context. Results demonstrate that models finetuned with LongCE achieve better long-context downstream task performance.

**Strengths:**

- This paper proposes a new metric LongPPL which better correlates with long-context downstream performance.
- This paper proposes a new training objective LongCE tailored for improving long-context performance.

**Weaknesses:**

- The proposed LongPPL is limited by the evaluator model. Concretely, the validity of the evaluation is strictly bounded by the maximum effective length that an evaluator model can reliably handle. Additionally, the two thresholds, which depend on both the evaluation data and the evaluator model, can further complicate the evaluation process. These limitations weaken the universality of the proposed metric.
- The key token selection of LongPPL is motivated by observing large gains in log probability of the ‘answer’ tokens in LongEval, a retrieval-based synthetic test. It raises the question of whether the gains primarily come from the improved prediction of less frequent n-grams that re-appear later in the context, as studied by [1][2]. If so, the motivation of leveraging an evaluator model is obscure since one can simply evaluate those re-occurred n-gram tokens, which are much more convenient to find and overall a more universal setup than LongPPL.
- The LongCE loss should be compared with the standard focal loss [3] baseline to understand if it is really necessary to add the contrasting term of long and short context perplexity. The standard focal loss does not require extra forward pass for computing short context log probability, thus can potentially improve training efficiency.
- The LongCE loss seems to improve mostly the retrieval-based test (LongEval), however, demonstrate little gains on RULER, which has a few non-retrieval tasks, and almost zero improvement on LongBench, which contains quite a few non-retrieval tasks (Figure 7). It is unclear to what extent the proposed method benefits non-retrieval tasks.


[1] Zoology: Measuring and Improving Recall in Efficient Language Models

[2] Do Long-Range Language Models Actually Use Long-Range Context?

[3] Focal Loss for Dense Object Detection

**Questions:**

- Why does finetuning with LongCE demonstrate such different results for mistral-7b and llama2 on RULER, with one hardly improving (<1) and one significantly improving (~10)?
- Why does figure 6 only contain correlation between longPPL and longbench score? What is the correlation between standard PPL vs longbench score?

---

> ### Author Response · Authors · 2024-11-20
> **Response to Reviewer At6Q (Part 1/3)**
>
> We thank reviewer At6Q for providing detailed comments, and we will address your concerns in the following points.
>
> ---
>
> **Q1.**
>
> > The proposed LongPPL is limited by the evaluator model. Concretely, the validity of the evaluation is strictly bounded by the maximum effective length that an evaluator model can reliably handle.
> >
>
> **A1.1.** Although the performance of LongPPL depends on the capability of the evaluator model, our experiments show that even an 8B-level model is also capable of handling it. Furthermore, since the choice of evaluator model is flexible, the boundaries of LongPPL’s capability will expand as the long-context capability of open-source models improves.
>
> > Additionally, the two thresholds, which depend on both the evaluation data and the evaluator model, can further complicate the evaluation process. These limitations weaken the universality of the proposed metric.
> >
>
> **A1.2.** We conduct an ablation study on the two thresholds in LongPPL (with Qwen2-72B-Instruct) in the following table. We find that when \beta=1, \alpha=1 or 2, the correlation between LongPPL and benchmarks even improves. In fact, we simply reused the hyperparameters summarized from the previous experiments without any overtuning. The results show that LongPPL’s performance is not sensitive to the choice of hyperparameters, with the correlation coefficient being greater than 0.8 in most cases.
>
> | Corr_coef | LongBench | LongEval | RULER |
> | --- | --- | --- | --- |
> | \alpha=2, \beta=2 (default) | -0.96 | -0.86 | -0.84 |
> | \alpha=2, \beta=1 | -0.96 | **-0.92** | **-0.92** |
> | \alpha=1, \beta=2 | -0.91 | -0.73 | -0.69 |
> | \alpha=1, \beta=1 | **-0.97** | -0.88 | -0.87 |
>
> We have included the above results in Appendix B.2.
>
> ---
>
> **Q2.** The key token selection of LongPPL is motivated by observing large gains in log probability of the ‘answer’ tokens in LongEval, a retrieval-based synthetic test. It raises the question of whether the gains primarily come from the improved prediction of less frequent n-grams that re-appear later in the context, as studied by [1][2]. If so, the motivation of leveraging an evaluator model is obscure since one can simply evaluate those re-occurred n-gram tokens, which are much more convenient to find and overall a more universal setup than LongPPL.
>
> **A2.** Following your suggestion, we calculate perplexity solely on the re-occurred N-gram (word-level, N>2) in the inputs. We present the correlation coefficients between the new perplexity metric and the benchmark scores in the table below.
>
> | Corr_coef | LongBench | LongEval | RULER |
> | --- | --- | --- | --- |
> | PPL | -0.11 | 0.24 | 0.27 |
> | LongEvalPPL (N-gram) | -0.41 | -0.10 | -0.05 |
> | LongPPL | **-0.96** | **-0.86** | **-0.84** |
>
> The results show that PPL on re-occurred N-grams has much weaker correlation with model’s long-context capabilities. This indicates that LongPPL’s powerful ability to capture long-context-related information cannot be simply explained by N-grams. However, the experimental results show that N-grams do have some effect, and we will include this discussion in the manuscript (Appendix B.6). Thank you for your constructive insight.
>
> ---
>
> **Q3.** The LongCE loss should be compared with the standard focal loss [3] baseline to understand if it is really necessary to add the contrasting term of long and short context perplexity. The standard focal loss does not require extra forward pass for computing short context log probability, thus can potentially improve training efficiency.
>
> **A3.** Following your suggestion, we finetune Llama-2-7B with FocalLoss ($\gamma$=2), and the results are shown below. It turns out that FocalLoss is inferior than LongCE, but performs better than CE in LongEval and RULER. This indicates that, although general re-weighting methods like FocalLoss can improve the model’s long-context capability to some extent, the re-weighting method based on long-short contrast in LongCE is more effective in long-context scenarios.
>
> | PG-19 dataset with EABF | LongBench |  |  | LongEval |  |  | RULER |  |  |
> | --- | --- | --- | --- | --- | --- | --- | --- | --- | --- |
> | Training Steps | 50 steps | 100 steps | 200 steps | 50 steps | 100 steps | 200 steps | 50 steps | 100 steps | 200 steps |
> | CE | 24.5 | 26.6 | 26.9 | 16.0 | 24.0 | 24.0 | 34.5 | 38.6 | 42.7 |
> | FocalLoss($\gamma$=2) | 24.2 | 23.7 | 24.4 | 14.0 | 22.0 | 30.0 | 37.8 | 43.2 | 45.1 |
> |  LongCE | **26.0** | **27.2** | **28.2** | **24.0** | **46.0** | **46.0** | **43.1** | **48.3** | **49.7** |
>
> ---

---

> ### Author Response · Authors · 2024-11-20
> **Response to Reviewer At6Q (Part 2/3)**
>
> **Q4.** The LongCE loss seems to improve mostly the retrieval-based test (LongEval), however, demonstrate little gains on RULER, which has a few non-retrieval tasks, and almost zero improvement on LongBench, which contains quite a few non-retrieval tasks (Figure 7). It is unclear to what extent the proposed method benefits non-retrieval tasks.
>
> **A4.** We present the detailed LongBench scores of our main experiments of LongCE in the following table. The results show that LongCE not only significantly outperforms CE on retrieval tasks (correspond to “synthetic tasks”), but also excels in summarization and QA tasks. On the contrary, LongCE lags behind in few-shot learning. The reason for this may be that, in few-shot learning tasks, the information in the prompt is denser, and the importance of each token tends to be more equal. In principal, LongCE is better suited for scenarios where the importance of prompt tokens is not equally distributed. The results are added in Appendix B.3.
>
> | Task Domains               | Single-Document QA | Multi-Document QA | Summarization | Few-shot Learning | Code Completion | Synthetic Tasks | Avg.  |
> |----------------------------|--------------------|--------------------|---------------|-------------------|-----------------|------------------|-------|
> | **Setting A (PG-19 dataset with EABF)**                                                                               |
> | CE (50 steps)              | 4.4                | 1.1                | 15.5          | 66.7              | 59.7            | 0.0              | 24.5  |
> | CE (100 steps)             | 5.9                | 2.0                | 21.9          | **67.5**          | 61.8            | 0.4              | 26.6  |
> | CE (200 steps)             | 6.9                | 2.3                | 22.8          | 66.8              | **61.9**        | 0.4              | 26.9  |
> | LongCE (50 steps)          | 7.6                | 2.1                | 22.0          | 66.1              | 57.9            | 0.5              | 26.0  |
> | LongCE (100 steps)         | 7.7                | 3.3                | 22.5          | 65.7              | 61.6            | 2.3              | 27.2  |
> | LongCE (200 steps)         | **9.3**            | **4.8**            | **23.9**      | 66.0              | **61.9**        | **3.2**          | **28.2** |
> | **Setting B (PG-19 dataset with PI)**                                                                                  |
> | CE (50 steps)              | 3.1                | 3.2                | 12.9          | 65.3              | 59.8            | 1.6              | 24.3  |
> | CE (100 steps)             | 4.1                | 3.5                | 17.5          | 65.2              | 59.9            | 1.8              | 25.3  |
> | CE (200 steps)             | 5.6                | 4.0                | 15.4          | **66.0**          | **60.3**        | 1.0              | 25.4  |
> | LongCE (50 steps)          | 4.5                | 2.2                | 15.6          | 63.1              | 58.4            | 2.7              | 24.4  |
> | LongCE (100 steps)         | 4.6                | 1.7                | 17.7          | 64.1              | 59.0            | **2.8**          | 25.0  |
> | LongCE (200 steps)         | **6.0**            | **4.3**            | **19.0**      | 63.6              | 59.2            | 2.7              | **25.8** |
> | **Setting C (Pile-arxiv dataset with EABF)**                                                                          |
> | CE (50 steps)              | 1.7                | 0.0                | 0.0           | 50.2              | 38.2            | 0.0              | 15.0  |
> | CE (100 steps)             | 4.2                | 5.4                | 4.9           | **65.0**              | 58.9            | 0.0              | 23.1  |
> | CE (200 steps)             | **5.1**            | **7.1**            | 7.6           | 64.3              | 58.7            | 0.0              | 23.8  |
> | LongCE (50 steps)          | 3.5                | 0.0                | 2.6           | 52.9              | 46.7            | 0.0              | 17.6  |
> | LongCE (100 steps)         | 4.2                | 5.3                | 10.0          | 64.3              | 59.1            | 1.0              | 24.0  |
> | LongCE (200 steps)         | 3.7                | 6.1                | **14.3**      | 64.7          | **59.8**        | **1.3**          | **25.0** |
>
> ---

---

> ### Author Response · Authors · 2024-11-20
> **Response to Reviewer At6Q (Part 3/3)**
>
> **Q5.** Why does finetuning with LongCE demonstrate such different results for mistral-7b and llama2 on RULER, with one hardly improving (<1) and one significantly improving (~10)?
>
> **A5.** Since LongCE uses the model itself as the evaluator, during the early stages of training, the quality of LongPPL re-weighting is closely related to the model itself. Therefore, the difference in performance between Mistral-7B and Llama2-13B is likely related to their initial differences in model capabilities.
>
> ---
>
> **Q6.** Why does figure 6 only contain correlation between longPPL and longbench score? What is the correlation between standard PPL vs longbench score?
>
> **A6.**  Figure 6 shows the correlation between LongPPL and LongBench score when calculated using an evaluator model different from that in Figure 1 (right). Standard PPL, however, does not change with different evaluator models. The correlation between PPL and LongBench is in Figure 1 (left). We now added the clarification in the caption of figure 6.
>
> Additionally, we have summarized their correlation coefficients in the table below, to further clarify the relationship between LongPPL, PPL, and the benchmarks. The results indicate that the correlation between LongPPL and the benchmark remains consistently high.
>
> | Corr_coef | LongPPL(Qwen-72B) | LongPPL(Mistral Large) | LongPPL(Llama-8B) | PPL |
> | --- | --- | --- | --- | --- |
> | LongBench | -0.96 | -0.79 | -0.96 | -0.11 |
> | LongEval | -0.86 | -0.91 | -0.89 | 0.24 |
> | RULER | -0.84 | -0.96 | -0.90 | 0.27 |
>
> ---
>
> Thanks again for your careful reading and detailed review. Please let us know if there is more to clarify. We are happy to take your further question in the discussion stage.

---

> > ### Comment · Reviewer_At6Q · 2024-11-22
> >
> > Thank you very much for the detailed replies. I will raise the score. The new results cleared up some of my concerns, while the limitation of having an evaluator model for computing LongPPL still remains.

---

> > > ### Author Response · Authors · 2024-11-23
> > >
> > > Thank you very much for raising the score! LongPPL’s ability to adaptively re-weight PPL without relying on manually designed benchmarks offers significant flexibility, making it a strong candidate as a long-context evaluation metric. Your constructive comments have been truly encouraging and are a driving force for our continuous improvement.

---

### Official Review · Reviewer_KLF8 · 2024-10-30

**Soundness:** 3
**Presentation:** 3
**Contribution:** 2
**Rating:** 6
**Confidence:** 2

**Summary:**

The authors state that perplexity metrics (PPL) are insufficient for evaluating the long-context capabilities of large language models (LLMs) because PPL considers both answer and non-answer tokens, resulting in perplexity scores that do not correlate with LongBench scores. To address this, the authors propose two new metrics, LongPPL and LongCE, which extract key tokens from the model output with the help of a relatively strong model as the evaluator. LongPPL is used as an evaluation metric for LLMs in long-context text, while LongCE serves as a loss function when fine-tuning LLMs on long-context tasks.

**Strengths:**

- The introduction of LongPPL and LongCE represents an innovation in the evaluation and fine-tuning of LLMs for long-context tasks. These metrics address the limitations of traditional perplexity and Cross Entropy loss, providing more accurate and task-relevant assessments of model performance.
- The concepts presented by the authors are easy to understand, including the motivation and method. The figures greatly help in understanding the concepts and the experiments.

**Weaknesses:**

- LongPPL metric relies on a relatively strong model (medium-sized Qwen2-72B-Instruct) as an evaluator to identify key tokens. This dependence may limit the applicability of the metric in scenarios where such a strong model is not available or practical to use.
- The additional computational cost associated with identifying key tokens may still be significant. This could be a barrier to adoption in resource-constrained environments.

**Questions:**

- How does the performance of LongPPL and LongCE change when using a weaker model for key token identification? Have you tried using the same model for both evaluation and key token identification?
- The paper discusses the overhead of using LongCE in LLM fine-tuning compared to the original Cross Entropy. How does the evaluation time of using LongPPL compare to traditional perplexity (PPL) metrics?

---

> ### Author Response · Authors · 2024-11-20
> **Response to Reviewer KLF8 (Part 1/2)**
>
> We thank Reviewer KLF8 for acknowledging the novelty and solidness of our work. Below, we address your main concerns.
>
> ---
>
> **Q1.** LongPPL metric relies on a relatively strong model (medium-sized Qwen2-72B-Instruct) as an evaluator to identify key tokens. This dependence may limit the applicability of the metric in scenarios where such a strong model is not available or practical to use.
>
> **A1.** We assume that by “strong” and “weak”, you are referring to the model size. Actually, LongPPL also supports using a smaller model as the evaluator model. In Figure 6, 8, 9, we use Llama-3.1-8B as the evaluator model to calculate LongPPL, and achieves high correlation as well.
>
> ---
>
> **Q2.** The additional computational cost associated with identifying key tokens may still be significant. This could be a barrier to adoption in resource-constrained environments.
>
> **A2.** We first introduce the theoretical complexity of LongPPL (LongCE), and then present their empirical overhead.
>
> Theoretically, the calculation of both LongPPL and LongCE involves two steps: (1) One forward pass on the full text, and (2) several forward passes on short texts (to obtain the LPG score for each token). Step 1, which involves calculations for both PPL (CE) and LongPPL (LongCE), has a time complexity of  $O(N^2)$ , where N is the input sequence length (N=32k in this paper). Step 2 represents the computational overhead of LongPPL (LongCE), with a time complexity of $O((N-K)K^2/d)$ , where K is the length of the short context(K=4k in this paper), and d is the sliding window length (d=1k in this paper). The detailed derivation can be found in Appendix A.1.
>
> Empirically, we conduct additional experiments to demonstrate the time cost of LongPPL and LongCE.
>
>
> ## LongPPL
>
> We test the time cost of LongPPL in the following table.
>
> | Evaluated model \ Time cost (second / seq) | PPL | LongPPL (Llama-3.1-8B) | LongPPL (Qwen2-72B) |
> | --- | --- | --- | --- |
> | Mistral-7B | 2.8 | 11.3 (+8.5, +304%) | 56.4 (+53.6, +2014%) |
> | Mixtral-8x7B (47B) | 4.2 | 13.5 (+9.3, +221%) | 58.4 (+54.2, +1390%) |
>
> It can be observed that the time cost of calculating LongPPL using the 8B model as the evaluator is approximately 3~4 times that of calculating PPL, while the overhead for using the 72B model is much higher.
>
> Although the computational overhead of LongPPL is non-negligible, we believe that such a computational cost will **not have a substantial impact on the practicality of LongPPL**. On the one hand, if users employ LongPPL as a benchmark, key tokens can be calculated offline, resulting in no online computation overhead. On the other hand, if LongPPL is used as an evaluation metric during training, its computational overhead is negligible compared to the overall training cost (as evaluation steps are typically sparse during training).
>
> We have included the above results in Appendix B.7.
>
> ## LongCE
> In section 4.2, we have already discussed the tradeoff between performance and computational cost of LongCE. To better illustrate this, we conducted an additional set of ablation experiments. In the following table, we present the long-context performance of models finetuned with LongCE under different values of K and d, which directly influence its computational efficiency.
>
> | PG-19 dataset with EABF | Total Training time (200 step) / h | LongBench |  |  | LongEval |  |  | RULER |  |  |
> | --- | --- | --- | --- | --- | --- | --- | --- | --- | --- | --- |
> |  |  | 50 steps | 100 steps | 200 steps | 50 steps | 100 steps | 200 steps | 50 steps | 100 steps | 200 steps |
> | K=4096, d=512 | 17.5 (+150%) | 25.4 | 25.8 | 27.3 | 26.0 | 48.0 | **60.0** | 42.4 | 50.1 | 54.4 |
> | K=4096, d=1024 (default) | 12.5 (+79%) | **26.0** | **27.2** | **28.2** | 24.0 | 46.0 | 46.0 | 43.1 | 48.3 | 49.7 |
> | K=4096, d=4096 | 9.5 (+36%) | 25.4 | 25.8 | 25.8 | **28.0** | **56.0** | 56.0 | 42.5 | 48.0 | 51.2 |
> | K=1024, d=1024 | 10.0 (+43%) | 25.3 | 25.8 | 26.9 | 20.0 | 48.0 | 48.0 | **45.6** | **51.1** | **55.9** |
> | CE | 7 | 24.5 | 26.6 | 26.9 | 16.0 | 24.0 | 24.0 | 34.5 | 38.6 | 42.7 |
>
> First, it can be observed that increasing K or decreasing d significantly improves the efficiency of LongCE (from +79% to +36%/+43%). Second, under these settings, although the model’s performance on real-world tasks (LongBench) decreases, it achieves substantial improvements on synthetic tasks (LongEval, RULER). This suggests that LongCE still holds potential for further efficiency enhancements. Therefore, we believe that the computational overhead of LongCE is entirely acceptable.
>
> We have included the above results in Appendix B.2.
>
> ---

---

> ### Author Response · Authors · 2024-11-20
> **Response to Reviewer KLF8 (Part 2/2)**
>
> **Q3.** How does the performance of LongPPL and LongCE change when using a weaker model for key token identification? Have you tried using the same model for both evaluation and key token identification?
>
> **A3.** For LongCE, we exactly use the same model as the evaluator (as explained in line 300). For LongPPL, in Table 5, we presented the LongPPL values when the evaluator model is the same as the model being evaluated, showing no correlation with long-context benchmarks.
>
> Intuitively, if the evaluated model is used to selects key tokens, then under the long-short contrastive mechanism, it can only identify key tokens that **it generates well,** missing those **it cannot generate well**. As a result, all models compute perplexity only on tokens they perform well on, which fails to reflect differences in the long-context capabilities of different models. To clarify, we have referenced the content of Table 5 in the main text of the revised version.
>
> ---
>
> **Q4.** The paper discusses the overhead of using LongCE in LLM fine-tuning compared to the original Cross Entropy. How does the evaluation time of using LongPPL compare to traditional perplexity (PPL) metrics?
>
> **A4.** Please see A2.
>
> ---
>
> Thanks again for your constructive suggestions. Hope our explanations can ease your concerns. Please let us know if there is more to clarify.

---

> ### Author Response · Authors · 2024-11-25
>
> Dear Reviewer KLF8,
>
> We have prepared a response to address your additional questions. Would you please take a look and let us know whether you find it satisfactory?
>
> Thank you! Wishing you a great day!
>
> The Authors

---

> > ### Comment · Reviewer_KLF8 · 2024-12-03
> > **Thank the authors for their response**
> >
> > Thank the authors for their response! I don't have any further questions.

---

### Official Review · Reviewer_kFZS · 2024-11-01

**Soundness:** 3
**Presentation:** 3
**Contribution:** 3
**Rating:** 6
**Confidence:** 4

**Summary:**

The paper critiques the use of perplexity (PPL) as a standard metric for evaluating large language models (LLMs) on long-context tasks, arguing that PPL’s averaging across all tokens fails to capture the importance of "key tokens" critical for understanding long-context dependencies. To address this, the authors introduce Log Probability Gain (LPG) and Log Probability Values (LPV) to identify these key tokens, leading to LongPPL—a refined metric that emphasizes these crucial tokens, thus providing a better measure of long-context performance. Additionally, they propose LongCE, a re-weighting strategy applied during fine-tuning that bolsters the model's ability to handle long-context tasks through an expectation-maximization (EM) approach. Although LongCE adds computational overhead, the authors show that it consistently outperforms traditional cross-entropy (CE) across various settings.

**Strengths:**

This paper conducts a fine-grained analysis to investigate why perplexity (PPL) fails to capture long-context capabilities in LLMs. Few previous studies focus on re-weighting tokens to improve long-context performance; this work addresses this directly by re-weighting tokens based on their dependence on long-context information, providing a more fundamental and targeted solution.

**Weaknesses:**

Some settings in the figures lack clear explanations, which affects the reliability of the figures and the conclusions drawn from them. For example, in Figure 2, the specific prompt lengths for each data point are not provided, making it difficult to interpret the statement, "Each point represents the results obtained from testing at a specific prompt length."
Although the correlation arguments seem promising and qualitatively sound, they would benefit from added statistical rigor, such as p-values from hypothesis testing to confirm the significance.
The naming of Log Probability Gain (LPG) and Log Probability Values (LPV) is not intuitive, which may hinder understanding and adoption of these concepts.
Additionally, the experiments are conducted with a sequence length of 32K, a length considered relatively manageable by recent research. Testing on longer sequences could introduce challenges that may not align with current trends, potentially impacting the generalizability of the results
Lastly, the criteria for selecting experimental models are not clearly defined, which introduces some ambiguity regarding the rationale behind the choice of models used in testing.

**Questions:**

1. Is the main reason perplexity fails due to its inability to correctly identify answer/key tokens?
2. Could you clarify what "task-aware" means in this context? Does it refer to distinctions between short-context and long-context tasks, or could it apply to other task definitions as well?
3. You mention that "even though the Llama-3.1 model we use is only 8B in size, it still obtains LongPPL with a strong correlation, suggesting high compatibility with powerful long-context models, even with smaller parameter sizes." Are you implying that LongPPL would show higher correlation with larger models? If so, could you explain why?

---

> ### Author Response · Authors · 2024-11-20
> **Response to Reviewer kFZS (Part 1/2)**
>
> We thank Reviewer kFZS for providing constructive questions for our work. We will address your concerns in the following points.
>
> ---
>
> **Q1.** Some settings in the figures lack clear explanations, which affects the reliability of the figures and the conclusions drawn from them. For example, in Figure 2, the specific prompt lengths for each data point are not provided, making it difficult to interpret the statement, "Each point represents the results obtained from testing at a specific prompt length."
>
> **A1.** Following your suggestion, we have added the detailed experimental data from Figure 2 in Appendix B.4, including the prompt length corresponding to each point and its associated LongEval Acc and PPL. Additionally, we have included “ranging from 2k to 28k” in the caption of Figure 2 to clarify the range of prompt lengths. We hope these can enhance the clarity of our experiments, and please let us know if there is more to clarify.
>
> ---
>
> **Q2.** Although the correlation arguments seem promising and qualitatively sound, they would benefit from added statistical rigor, such as p-values from hypothesis testing to confirm the significance.
>
> **A2.** Following your suggestion, we have added the p-values in Figure 1, 5, 6, 8, 9. We summarize them in the following table. The results show that the p-values between LongPPL and the benchmarks are all less than or equal to 0.01, indicating **a significant linear correlation**. In contrast, the p-values between PPL and the Benchmark are all greater than or equal to 0.49, indicating no significant linear correlation. These results are consistent with the trends reflected by the correlation coefficients.
>
> | P-value | LongPPL(Qwen-72B) | LongPPL(Mistral Large) | LongPPL(Llama-8B) | PPL |
> | --- | --- | --- | --- | --- |
> | LongBench | $4\times10^{-5}$ | 0.01 | $4\times10^{-5}$ | 0.64 |
> | LongEval | 0.002 | 0.001 | 0.001 | 0.54 |
> | RULER | 0.005 | $4\times10^{-5}$ | 0.001 | 0.49 |
>
> ---
>
> **Q3.** The naming of Log Probability Gain (LPG) and Log Probability Values (LPV) is not intuitive, which may hinder understanding and adoption of these concepts.
>
> **A3.** Following your suggestion, we change the naming of LPG and LPV to Long-Short Difference (LSD) and Long-Context Likelihood (LCL), respectively. Hope this will make these concepts easier to understand.
>
> ---
>
> **Q4.** Additionally, the experiments are conducted with a sequence length of 32K, a length considered relatively manageable by recent research. Testing on longer sequences could introduce challenges that may not align with current trends, potentially impacting the generalizability of the results.
>
> **A4.** To evaluate the long-context capabilities over 32k, we conduct the standard Needle-in-a-Haystack (NIAH) evaluation in Appendix B.5. We first test the models obtained in our main text, which are fine-tuned on 32K-length texts. As shown in figure 11, LongCE achieves a score of 10 on 5 out of 6 questions at the 40K length and 2 out of 6 questions at the 48K length, outperforming CE, which achieves a score of 10 on 2 out of 6 and 0 out of 6 questions, respectively. Therefore, LongCE demonstrates a longer effective context length.
>
> Additionally, to demonstrate the generalization ability of LongCE on longer context lengths, we extend the context window of both models by increasing their RoPE base from 500K to 2M. The corresponding NIAH results are shown in Figure 12. The results show that model finetuned with LongCE answers all questions correctly at the 64K length and achieves a score of 10 on 32 sequences with lengths of ≥32K, while CE only achieves this on 26 sequences. This indicates that LongCE can generalize well at longer lengths.
>
> ---
>
> **Q5.** Lastly, the criteria for selecting experimental models are not clearly defined, which introduces some ambiguity regarding the rationale behind the choice of models used in testing.
>
> **A5.** We are not sure whether you are referring to LongPPL or LongCE, so we will address your questions separately for each.
>
> - **LongPPL.** As shown in Appendix D, the models we evaluated cover most of the mainstream long-context models available today. The vast majority of them, such as Qwen, Mistral (Mixtral), Llama, Yi, Phi, FILM, and Yarn, are long-context LLMs listed on RULER. These models encompass different model families as well as various long-context extension techniques (e.g., modification on RoPE, long-context fine-tuning, sliding window attention, etc.).
> - **LongCE.** In Table 2, we use the widely used Llama-2-7B as our base model. In Table 3, we first use Mistral-7B as the base model to show that LongCE is effective across different model families. We then use Llama-2-13B to demonstrate that LongCE is also effective across different model sizes.
>
> ---

---

> ### Author Response · Authors · 2024-11-20
> **Response to Reviewer kFZS (Part 2/2)**
>
> **Q6.** Is the main reason perplexity fails due to its inability to correctly identify answer/key tokens?
>
> **A6.** Yes. Since PPL cannot identify key tokens, it treats all tokens equally. However, because key tokens only constitute a small portion of natural text, PPL does not give them enough attention, and thus fails to reflect the ability to handle long texts.
>
> ---
>
> **Q7.** Could you clarify what "task-aware" means in this context? Does it refer to distinctions between short-context and long-context tasks, or could it apply to other task definitions as well?
>
> **A7.** In the beginning of section 3.1, we mention that the FocalPPL (Eq. 4) is *task-aware* by modifying the influence function I(x). The term “task” in this article refers to long-contexts, but it can also represent other general tasks (such as toxicity, safety). Here, we use the notion of FocalLoss to make the topic more general, and we will leave these other tasks for future work.
>
> ---
>
> **Q8.** You mention that "even though the Llama-3.1 model we use is only 8B in size, it still obtains LongPPL with a strong correlation, suggesting high compatibility with powerful long-context models, even with smaller parameter sizes." Are you implying that LongPPL would show higher correlation with larger models? If so, could you explain why?
>
> **A8.** We initially thought that an 8B-level model would not be suitable as an evaluator model, but experiments have shown that it works well. Therefore, the statement you mentioned is actually saying that Llama-3.1-8B works well, rather than suggesting that LongPPL would show higher correlation with larger models.
>
> ---
>
> Thanks for your careful reading and detailed review. Please let us know if there is more to clarify. We are happy to take your further question in the discussion stage.

---

> ### Author Response · Authors · 2024-11-25
>
> Dear Reviewer kFZS,
>
> We have prepared a response to address your additional questions. Would you please take a look and let us know whether you find it satisfactory?
>
> Thank you! Wishing you a great day!
>
> The Authors

---

### Official Review · Reviewer_aS6q · 2024-11-09

**Soundness:** 3
**Presentation:** 3
**Contribution:** 3
**Rating:** 8
**Confidence:** 3

**Summary:**

This paper modified standard PPL and CE (cross entropy) loss, by weighting more to key tokens in long text. The key tokens are identified through a long-short context constrastive method. The experiments demonstrate the LongPPL correlates long-context capabilities of LLM, thus a good metric. Similarly, applying LongCE loss in long-context fine tuning shows improvement on long-context benchmarks.

**Strengths:**

The idea is simple and seems working well.

**Weaknesses:**

n/a

**Questions:**

1. I wonder how would fine tuned model perform on normal context length benchmarks, will it remain roughly at the same level or regress a bit?
2. How long is considered as "long-context". In experiments, 32k is used. I wonder how much this method would affect on other context lengths.
3. can we use equation (5) directly for LongCE instead of (8). so only differentiate on key tokens, would that be doable?

---

> ### Author Response · Authors · 2024-11-20
> **Response to Reviewer aS6q (Part 1/2)**
>
> We thank reviewer aS6q for appreciating the idea of our paper, and we will address your concerns in the following points.
>
> ---
>
> **Q1.** I wonder how would fine-tuned model perform on normal context length benchmarks, will it remain roughly at the same level or regress a bit?
>
> **A1.** Please see A2 for experimental results on normal context length.
>
> ---
>
> **Q2.** How long is considered as "long-context". In experiments, 32k is used. I wonder how much this method would affect on other context lengths.
>
> **A2.** Based on our observations of recent research, the term “long context” primarily refers to context lengths that significantly exceeding the 4K length of models like Llama-2-7B. Although recent research has extended text lengths to 64K [1], 128K [2-3], and even longer [4], 32K is still widely accepted as the standard “long context” setting in this field [5-8].
>
> To further address your concerns about the generalizability of LongCE to other context lengths, we conducted the following experiments.
>
> **Normal context length (4k/8k).** First, we present the performance of models fine-tuned with CE and LongCE on normal context length (4k/8k), as shown in the table below (”0 step“ refers to the vanilla Llama-2-7B). In the default setting in our paper, we used a context length of 4096 for computing the short-context likelihood $P(x_i|s_i)$ in the LongCE loss, aiming at extending to a context longer than 4096. Here, to improve the performance at 4096 as well, we use a shorter context length of 1024 for computing $P(x_i|s_i)$ instead.
>
> | PG-19 dataset with EABF | Context Length |  | LongBench |  |  |  | LongEval |  |  |  | RULER |  |  |
> | --- | --- | --- | --- | --- | --- | --- | --- | --- | --- | --- | --- | --- | --- |
> | Training Steps |  | 0 step | 50 steps | 100 steps | 200 steps | 0 step | 50 steps | 100 steps | 200 steps | 0 step | 50 steps | 100 steps | 200 steps |
> | CE | 4k | 26.5 | 23.8 | **25.5** | 25.9 | 86.0 | 82.0 | 86.0 | 82.0 | 85.6 | **57.7** | **68.2** | **73.2** |
> | LongCE |  |  | **24.6** | **25.5** | **26.4** |  | **92.0** | **98.0** | **100.0** |  | **69.7** | **72.6** | **75.5** |
> | CE | 8k | - | 24.2 | **26.2** | 26.4 | - | 58.0 | 84.0 | 78.0 | - | 48.8 | 58.0 | 63.5 |
> | LongCE |  |  | **25.6** | 26.0 | **26.7** |  | **78.0** | **98.0** | **100.0** |  | **62.0** | **65.0** | **68.3** |
> | CE | 32k | - | 24.5 | **26.6** | **26.9** | - | 16.0 | 24.0 | 24.0 | - | 34.5 | 38.6 | 42.7 |
> | LongCE |  |  | **25.3** | 25.8 | **26.9** |  | **20.0** | **48.0** | **48.0** |  | **45.6** | **51.1** | **55.9** |
>
> Experimental results show that LongCE achieves consistent performance improvements over CE in both normal-length and long contexts. This indicates that, by shortening the length of the short context in LongCE, the model achieves a balance between its capabilities on both long context and normal-length context.
>
> **Longer context lengths(>32k).** Second, we conduct the standard Needle-in-a-Haystack (NIAH) evaluation in Appendix B.5 to evaluate models’ long-context capability when context lengths is greater than 32K. We first test the models obtained in our main text, which are fine-tuned on 32K-length texts. As shown in figure 11, LongCE achieves a score of 10 on 5 out of 6 questions at the 40K length and 2 out of 6 questions at the 48K length, outperforming CE, which achieves a score of 10 on 2 out of 6 and 0 out of 6 questions, respectively. Therefore, LongCE demonstrates a longer effective context length.
>
> Additionally, to demonstrate the generalization ability of LongCE on longer context lengths, we extend the context window of both models by increasing their RoPE base from 500K to 2M. The corresponding NIAH results are shown in Figure 12. The results show that model finetuned with LongCE answers all questions correctly at the 64K length and achieves a score of 10 on 32 sequences with lengths of ≥32K, while CE only achieves this on 26 sequences. This indicates that LongCE can generalize well at longer lengths.
>
> ---

---

> ### Author Response · Authors · 2024-11-20
> **Response to Reviewer aS6q (Part 2/2)**
>
> **Q3.** Can we use equation (5) directly for LongCE instead of (8). so only differentiate on key tokens, would that be doable?
>
> **A3.** Following your suggestion, we finetuned Llama-2-7B with LongCE using equation (5) as the reweighting function. The results are given in the table below.
>
> | PG-19 dataset with EABF | LongBench |  |  | LongEval  |  |  | RULER  |  |  |
> | --- | --- | --- | --- | --- | --- | --- | --- | --- | --- |
> | Training Steps | 50ep | 100ep | 200ep | 50ep | 100ep | 200ep | 50ep | 100ep | 200ep |
> | LongCE (hard metric, Eq. 5) | 24.8 | **27.3** | 25.5 | 16.0 | 40.0 | 40.0 | 39.8 | 47.0 | 48.3 |
> | LongCE (default, Eq. 8) | **26.0** | 27.2 | **28.2** | **24.0** | **46.0** | **46.0** | **43.1** | **48.3** | **49.7** |
>
> The results show that when directly using the hard criteria in Eq.5 to re-weight LongCE, i.e., calculating gradients only on key tokens, the model’s performance is inferior than using the soft criteria in Eq.8. In fact, the key tokens selected using hard criteria typically account for only a small proportion (approximately 1–2%). Training exclusively on these tokens would result in very low data utilization efficiency. In contrast, methods like LongRecipe [2], which also focus on a subset of tokens for long-context training, use at least 30% of the tokens. Therefore, it is more reasonable to use soft criteria to weight the tokens.
>
> [1] Chen, Guanzheng, et al. "Clex: Continuous length extrapolation for large language models." *arXiv preprint arXiv:2310.16450* (2023).
>
> [2] Hu, Zhiyuan, et al. "Longrecipe: Recipe for efficient long context generalization in large languge models." *arXiv preprint arXiv:2409.00509* (2024).
>
> [3]Peng, Bowen, et al. "Yarn: Efficient context window extension of large language models." *arXiv preprint arXiv:2309.00071* (2023).
>
> [4] Young, Alex, et al. "Yi: Open foundation models by 01. ai." *arXiv preprint arXiv:2403.04652* (2024).
>
> [5] Zhang, Yikai, Junlong Li, and Pengfei Liu. "Extending LLMs' Context Window with 100 Samples." *arXiv preprint arXiv:2401.07004* (2024).
>
> [6] An, Shengnan, et al. "Make Your LLM Fully Utilize the Context." *arXiv preprint arXiv:2404.16811* (2024).
>
> [7] Wang, Suyuchen, et al. "Resonance rope: Improving context length generalization of large language models." *arXiv preprint arXiv:2403.00071* (2024).
>
> [8] Chen, Shouyuan, et al. "Extending context window of large language models via positional interpolation." *arXiv preprint arXiv:2306.15595* (2023).
>
> ---
>
> Thanks again for your constructive suggestions. Please let us know if there is more to clarify.

---

### Author Response · Authors · 2024-11-20
**Updates in the manuscript**

Following the suggestion from the reviewers, we have updated a revised manuscript with the main changes highlighted in orange, which are:

1. To make the concepts easier to understand, we changed the expression of "Log Probability Gain (LPG)" and "Log Probability Value (LPV)" to "Long-Short Difference (LSD)" and "Long-Context Likelihood (LCL)", respectively.

2. To improve clarity, we added statements in the section 4 of main text to reference the experimental results.

3. We added additional discussions and ablation experiments in the appendix in response to the reviewers’ feedback.

- We added discussion about the complexity of LongPPL (LongCE) in Appendix A.1.

- We added ablation study about the hyperparameters in LongPPL ($\alpha$, $\beta$) in Appendix B.2. The results show that LongPPL is not sensitive to the hyperparameters.

- We added ablation study about the hyperparameters in LongCE ($K$, $d$) in Appendix B.2. The results indicate that by selecting different hyperparameters, LongCE can achieve a balance between performance and efficiency.

- We provided fine-grained LongBench scores of LongCE in Appendix B.3, showing that LongCE mainly outperforms in retrieval, summarization and QA tasks.

- We provided detailed results from the experiments in Figure 2 in Appendix B.4 to improve clarity.

- We added needle-in-a-haystack test in Appendix B.5, to show how models fine-tuned with LongCE generalize on context length > 32k.

- We conducted experiments to substitute the key tokens of LongPPL with re-occurred N-gram in Appendix B.6. The experiments show that using N-grams as key tokens fails to achieve the effectiveness of LongPPL, indicating that N-grams cannot fully explain LongPPL’s mechanism for selecting key tokens.

- We provided the time cost of LongPPL in Appendix B.7 and explain that the time cost is managable.

---

### Meta-Review · Area_Chair_5TJr · 2024-12-17

**Metareview:**

This paper addresses the limitations of perplexity (PPL) as a metric for evaluating the long-context capabilities of large language models (LLMs). Standard PPL averages over all tokens, failing to highlight the role of "key tokens" essential for understanding long-context dependencies. The authors further propose two innovations to address the limitations of PPL. 1) LongPPL: A refined evaluation metric that focuses on key tokens, which are identified using a long-short context contrastive approach. The approach computes the difference in token probabilities between long and short contexts (Log Probability Gain, LPG). Tokens with high LPG scores are considered key. LongPPL demonstrates stronger correlation with long-context benchmarks (e.g., LongBench) compared to standard PPL. 2) LongCE: A re-weighted cross-entropy loss function for fine-tuning LLMs, to give key tokens higher importance during training, dynamically identified based on their significance in long-context scenarios. Inspired by focal loss, LongCE adapts weights to improve long-context understanding. Experiments show that models fine-tuned with LongCE outperform those trained with traditional cross-entropy loss on long-context benchmarks. The paper proposes a well-motivated and innovative approach to tackle the shortcomings of standard metrics and training strategies for long-context tasks in LLMs. Its clear methodology, strong empirical results, and practical relevance make it a significant contribution to the field. However, further exploration of computational efficiency and broader validation would enhance its impact.

Strength of this paper

- Novel Contributions: Introduces LongPPL, a new evaluation metric that correlates more effectively with long-context downstream performance than standard perplexity (PPL). Proposes LongCE, a novel training objective tailored to improve long-context performance by dynamically re-weighting tokens based on their dependence on long-context information. The identification of key tokens and their use in both metrics and training is a simple yet effective approach.
- Addressing Fundamental Limitations: Conducts a detailed analysis of why standard PPL fails to capture long-context capabilities in LLMs, providing a well-motivated and targeted solution. Offers a convincing explanation, supported by illustrations, that highlights the inadequacy of traditional metrics for long-context tasks.
- The paper is well-structured and progressive, starting from problem identification, explaining its causes, and presenting efficient solutions.
- Extensive experiments validate the efficacy of proposed metrics and training objectives, showing consistent improvements across diverse benchmarks.
- Impactful Findings: the work tackles an under-explored area of long-context performance in LLMs, offering a fundamental rethinking of token weighting and evaluation metrics, and provides tools and strategies that can be broadly applied to improve long-context tasks.


Weakness of this paper:

Several reviewers raised few concerns/limitations of this paper. By addressing these limitations, the paper could strengthen its experiment and expand impact.

- Computational Overhead: LongCE introduces additional computation by dynamically identifying and prioritizing key tokens, and requires additional forward passes for short-context log probability. Both  could be barriers in resource-constrained settings. Some analyses on computational costs would be helpful. Besides, simpler methods to avoid using evaluator model for key token selection , such as evaluating re-occurring n-grams, might offer similar benefits without the complexity.
- Dependency on Evaluator Models: LongPPL relies on an external evaluator to identify key tokens, which may add complexity to implementation. The validity of LongPPL is also bounded by the evaluator model's maximum effective length and thresholds, complicating its universal application.
- Experimental Limitations: The experiments use a sequence length of 32K, which is relatively manageable with current technologies. Testing on longer sequences could reveal additional challenges and generalizability issues. LongCE demonstrates limited improvements on non-retrieval benchmarks like RULER and LongBench, raising questions about its broader applicability beyond retrieval-based tasks. Besides, The LongCE loss is not compared against standard focal loss, which could provide insights into whether the proposed contrasting term of long- and short-context perplexity is necessary.
- Lastly, the claim that perplexity (PPL) remains the de facto metric for evaluating long-context LLMs is debatable, as many recent works rely on benchmarks or specific tests like the needle-in-the-haystack test. Impact claimed by the authors might be over-estimated.

**Additional Comments On Reviewer Discussion:**

Above summarized the strength and weaknesses raised by reviewers. Most of the weaknesses were addressed via further discussion and more experiment results. Given the relatively positive ratings, the strengthens summarized above, and mitigated concern on weaknesses, I recommend to accept this paper.

---

### Decision · Program_Chairs · 2025-01-22

Accept (Poster)